# A genome-scale metabolic model of *Cupriavidus necator* H16 integrated with TraDIS and transcriptomic data reveals metabolic insights for biotechnological applications

**Nicole Pearcy**[1☯]*, **Marco Garavaglia**[1☯]*, **Thomas Millat**[1], **James P. Gilbert**[1¤a], **Yoseb Song**[2], **Hassan Hartman**[3¤b], **Craig Woods**[1¤c], **Claudio Tomi-Andrino**[1], **Rajesh Reddy Bommareddy**[1¤d], **Byung-Kwan Cho**[2,4], **David A. Fell**[3], **Mark Poolman**[3], **John R. King**[5], **Klaus Winzer**[1], **Jamie Twycross**[6]*, **Nigel P. Minton**[1]*

1 School of Life Sciences, University of Nottingham, Nottingham, United Kingdom, 2 Department of Biological Sciences, Korea Advanced Institute of Science and Technology, Daejeon, Republic of Korea, 3 School of Biological & Medical Sciences, Oxford Brookes University, Oxford, United Kingdom, 4 KAIST Institute for the BioCentury, Korea Advanced Institute of Science and Technology, Daejeon, Republic of Korea, 5 School of Mathematical Sciences, University of Nottingham, Nottingham, United Kingdom, 6 School of Computer Science, University of Nottingham, Nottingham, United Kingdom

☯ These authors contributed equally to this work.
¤a Current address: Janssen Research & Development, LLC., Observational Health and Data Analytics, Raritan, New Jersey, United States of America
¤b Current address: Joint Modelling Team, UK Health Security Agency, London, United Kingdom
¤c Current address: Deep Branch Biotechnology, MTIF Building, Clifton Lane, Nottingham, United Kingdom
¤d Current address: The hub for biotechnology in the built environment (HBBE), Department of Applied Sciences, Northumbria University, Newcastle upon Tyne, United Kingdom
* Nicole.pearcy@nottingham.ac.uk (NP); m.garavaglia@nottingham.ac.uk (MG); jamie.twycross@nottingham.ac.uk (JT); nigel.minton@nottingham.ac.uk (NPM)

**Data Availability Statement:** The raw FASTQ files generated from Illumina sequencing of the TraDIS libraries can be accessed from the NCBI website,

## Abstract

Exploiting biological processes to recycle renewable carbon into high value platform chemicals provides a sustainable and greener alternative to current reliance on petrochemicals. In this regard *Cupriavidus necator* H16 represents a particularly promising microbial chassis due to its ability to grow on a wide range of low-cost feedstocks, including the waste gas carbon dioxide, whilst also naturally producing large quantities of polyhydroxybutyrate (PHB) during nutrient-limited conditions. Understanding the complex metabolic behaviour of this bacterium is a prerequisite for the design of successful engineering strategies for optimising product yields. We present a genome-scale metabolic model (GSM) of *C. necator* H16 (denoted *i*CN1361), which is directly constructed from the BioCyc database to improve the readability and reusability of the model. After the initial automated construction, we have performed extensive curation and both theoretical and experimental validation. By carrying out a genome-wide essentiality screening using a Transposon-directed Insertion site Sequencing (TraDIS) approach, we showed that the model could predict gene knockout phenotypes with a high level of accuracy. Importantly, we indicate how experimental and computational predictions can be used to improve model structure and, thus, model

with BioProject ID PRJNAA833237. The SBML and ScrumPy version of the *i*CN1361 GSM are available from: https://github.com/SBRCNottingham/CnecatorGSM/tree/main/Model. The Jupyter Notebooks used to carry out the GSM analysis are also available from: https://github.com/SBRCNottingham/CnecatorGSM/tree/main/JupyterNotebooks.

**Funding:** This work was supported by the Biotechnology and Biological Sciences Research Council (BBSRC); grant number BB/L013940/1 and the Engineering and Physical Sciences Research Council (EPSRC) under the same grant number. The funders had no role in study design, data collection and analysis, decision to publish, or preparation of the manuscript.

**Competing interests:** The authors have declared that no competing interests exist.

accuracy as well as to evaluate potential false positives identified in the experiments. Finally, by integrating transcriptomics data with *i*CN1361 we create a condition-specific model, which, importantly, better reflects PHB production in *C. necator* H16. Observed changes in the omics data and *in-silico*-estimated alterations in fluxes were then used to predict the regulatory control of key cellular processes. The results presented demonstrate that *i*CN1361 is a valuable tool for unravelling the system-level metabolic behaviour of *C. necator* H16 and can provide useful insights for designing metabolic engineering strategies.

## Author summary

Genome-scale metabolic models (GSMs) provide a tool for unravelling the complex metabolic behaviour of bacteria and how they adapt to changing environments and genetic perturbations, and thus offer invaluable insights for biotechnology applications. For a GSM to be used efficiently for strain development purposes, however, the model must be easily readable and reusable by other researchers, whilst being able to predict metabolic behaviour with a high level of accuracy. In this work, we developed a GSM for *Cupriavidus necator* H16 that is linked to the BioCyc database, which provides an efficient way of application, model update, integration of experimental data and network visualisation for other researchers. Using our model, we demonstrate how integrating experimental observations, including Transposon-directed Insertion site Sequencing (TraDIS) and omics data, can be used to compensate for the lack of regulatory, kinetic and thermodynamic information in GSMs, and thus improve model accuracy. Importantly, we found that TraDIS *in vivo* screening and GSM analysis are complementary approaches, which can be used in combination to provide reliable gene essentiality predictions. Overall, our results offer an informed strategy for the deliberate manipulation of *C. necator* H16 metabolic capabilities, towards its industrial application to convert greenhouse gases into biochemicals and biofuels.

## Introduction

The development of alternative and sustainable routes for producing chemicals and fuels is one of the major challenges of the 21$^{st}$ century, due to the diminishing supply of fossil fuels and their severe damaging impact on the environment and human health through pollution and global warming [1]. Exploiting microbes as cellular factories converting renewable feedstocks into biomaterials, biochemicals and biofuels has thus attracted both academic and industrial interest as an alternative to fossil fuels [2]. Recent advances in metabolic engineering and synthetic biology tools have enabled genetic manipulation of selected microbial chassis to redirect carbon towards native and heterologous pathways for optimising the production and properties of desirable chemicals [3].

*Cupriavidus necator* H16 (previously known as *Alcaligenes eutrophus* and *Ralstonia eutropha*) is amongst the most attractive species to engineer as a microbial factory for producing bulk chemicals, due to its highly flexible metabolism, ability to grow to high cell densities and its genetic tractability [4–6]. Interestingly, the bacterium is capable of growing on an extremely wide variety of substrates, including sugars, fatty acids and aromatic compounds [7–9]. Of particular interest, however, is *C. necator*'s ability to utilise carbon dioxide ($CO_2$) as its sole carbon source, whilst utilising hydrogen ($H_2$) as its energy and electron source [10]. $CO_2$ is highly

abundant in waste off-gas from many industrial processes, such as steel and concrete manufacture, as well as energy generation, and is thus a major contributor to air pollution and global warming [11]. Metabolically engineering *C. necator* H16 to valorise $CO_2$ into chemicals and biofuels, thus has become an attractive next generation solution that simultaneously mitigates climate change and reduces our reliance on fossil fuels without competing with food resources [4], provided 'green' hydrogen is used.

Of further interest to industry, is the bacterium's natural ability to produce large quantities of polyhydroxyalkanoates (PHAs) as storage compounds during nutrient-limited conditions, such as nitrogen or oxygen, when carbon is readily available [7,12–15]. Importantly, PHAs are being considered as an alternative to the petrol-based thermoplastics due to their comparable material properties and biodegradability [16,17]. The current limitation for commercial use of PHAs, however, is the high cost of their synthesis compared to petrol-based plastics [18,19]. Exploiting *C. necator* H16 to produce PHAs, whilst also growing on cheap feedstocks, may therefore provide an economically viable and greener solution [15].

To successfully employ *C. necator* H16 as a chassis for the production of platform chemicals, however, requires a greater understanding of the bacterium's metabolic responses to perturbations [20]. Genome-scale metabolic models (GSMs) representing an organism's metabolic capabilities are an invaluable computational tool for such an endeavour. With the advent of whole genome sequencing and high-throughput data of all levels of biological organisation, their construction, parametrisation, and validation have been increasingly improved and initiated the development of advanced methods for their application [21,22].

Despite their key role in biotechnological applications, many GSMs, including the previously published model of *C. necator* H16 (named RehMBEL1391) [9], are constructed using non-conventional identifiers for reactions and metabolites, which limits their reusability and further development. Manually mapping identifiers in GSMs is extremely time-consuming for large-scale models and applying naive string comparisons can often lead to inconsistencies that invalidate curated models [23]. Notably, the authors of [24] have made some significant improvements to the original RehMBEL1391 model in their recent publication, including the addition of identifiers for a subset of metabolites and reactions. Stoichiometric and mass inconsistencies, however, remain in this updated model, which are challenging to correct without the manual curation of metabolite and reaction information due to the incomplete coverage of database-linked identifiers.

In this work, we therefore present a GSM that is directly constructed from the BioCyc Pathway Genome DataBase (PGDB) [25] for *C. necator* H16 using the ScrumPy software [26]. The benefits of constructing a GSM using this approach is 3-fold. First, the GSM includes metabolite and reaction identifiers that match the BioCyc database and thus enables accessibility to the plethora of tools available in BioCyc, including visualisation (such as metabolic pathways and metabolite chemical structures), omics data integration and comparisons to other organisms. Additionally, the BioCyc identifiers also provide greater accessibility to additional resources using their links to external databases, such as KEGG [27] and BRENDA [28]. Second, the model can be frequently updated as information in BioCyc is updated and improved. Third, unlike the standard SBML models, the ScrumPy model uses a modular approach that enables the capture of changes made during the model's development lifetime, as well as separately defined metabolic subsystems, improving the model's readability.

Furthermore, we tested the model's ability to predict gene-knockout phenotypes by employing a genome-wide essentiality screening using a TraDIS approach [29]. In addition, we used our new GSM to investigate the system-level metabolic changes that occur in *C. necator* H16 during nitrogen-limited conditions, and importantly, provide new insights into the alterations of metabolic fluxes under these conditions.

## Results

### Construction of the *C. necator* H16 genome-scale model

For the construction of our GSM of *C. necator* H16 we have used the semi-automated pipeline outlined in [30–34]. It applies a modular framework for GSM construction using the ScrumPy metabolic modelling software package [26]. Employing this approach, we developed a model that is divided into the following 7 submodules comprised of reactions derived either automatically (using ScrumPy) or manually:

- *AutoReutro*. A draft set of reactions automatically extracted from the BioCyc Pathway Genome Database (PGDB) (Reutro, v. 21.0) [25].

- *Transporters*. A set of reactions for the import and export of metabolites known to be taken up and/or secreted.

- *ETC*. Manually derived reactions related to the electron transport chain.

- *Biomass*. A lumped biomass equation representing a pseudo-stoichiometric reaction that consumes all essential precursors in the molar proportions, required for producing 1 gram dry cell weight.

- *PLS*. Reactions involved in the biosynthesis of phospholipids manually curated to consume specific fatty acids.

- *LPS*. Reactions involved in the lipopolysaccharides biosynthesis pathway manually curated to consume specific fatty acids.

- *ExtraReacs*. Manually added reactions based on gap filling for known metabolic capabilities, which were not included in BioCyc.

A top-level module (MetaReutro) then combines all submodules into the final GSM. This approach provides a convenient way of managing the development and curation of GSMs. The MetaReutro model (hereafter referred to as *i*CN1361) has a total of 1,292 reactions (98 of which are transporters) and 1,265 metabolites. Importantly, *i*CN1361 fully obeys the law of mass conservation for carbon, nitrogen, sulfur, oxygen, phosphate and hydrogen (including protons), and is also free from erroneous energy-generating cycles, which result from incorrect reaction directions or atom imbalances (see Methods for details of the theoretical validation tests and the MEMOTE report: https://github.com/SBRCNottingham/CnecatorGSM/blob/main/MEMOTE_repo/sbrc_cnecator_gsm/iCN1361_MEMOTE_report.html).

Next, to enable the model to predict gene knockouts and to allow integration of gene expression data, we derived the relationships between genes and reactions (*i.e.*, the combination(s) of genes required for the enzyme activity). Here, ScrumPy was again used to extract the gene(s) information associated with each reaction. The gene-reaction relationship was defined as a protein complex if multiple genes from the same operon were associated to the same reaction. We could assign a gene-reaction rule to 1,085 of the internal reactions in the model. The remaining reactions consisted of 15 spontaneous reactions and 94 reactions that had been added due to gap filling but for which no homologous gene could be found with the correct functional annotation.

The final version of the model is provided in both ScrumPy and SBML formats in the GitHub repository (https://github.com/SBRCNottingham/CnecatorGSM/tree/main/Model). An Excel file listing the metabolites, reactions and gene-relationships and biomass equation is also provided in the supplementary material (Tables A-E in S1 Data).

**Table 1. Network property comparisons between *i*CN1361 and RehMBEL1391.**

| Property | *i*CN1361 | RehMBEL1391 [24] |
|---|---|---|
| Number of reactions | 1292 | 1538 |
| Number of transporters | 98 | 384 |
| Number of internal reactions | 1194 | 1154 |
| Number of metabolites (cytosol) | 1263 | 1172 |
| Number of genes | 1361 | 1345 |
| Reactions with GPR* | 1110 | 1051 |
| Blocked reactions | 443 | 593 |
| Functional reactions | 793 | 738 |
| Balanced reactions (%) | 99.67 | 49.23 |
| Erroneous energy-generating cycles | 0 | 0 |
| Unconserved metabolites | 0 | 235 |
| Stoichiometric consistency | Yes | No |

*GPR: Gene-protein-reaction relationships (see the Methods section for a detailed explanation)

## Comparison to RehMBEL1391

We have carried out a comparison of the network properties of *i*CN1361 to the recently updated version of RehMBEL1391 GSM of *C. necator* H16 (Table 1). It should be noted that the significantly higher number of reactions in the recently updated version of RehMBEL1391 is mostly caused by a considerably larger number of reactions associated with transport. Only 17 of these transport reactions have been assigned a gene association, and thus it is unclear whether *C. necator* H16 is able to carry out the function. In *i*CN1361, we instead include only transporters where experimental evidence confirms the function is present. In all other categories listed in Table 1, our model *i*CN1361 scores higher than the updated version of RehMBEL1391. The MEMOTE report for RehMBEL1391 shows that, in strong contrast to *i*CN1361, stoichiometric inconsistencies and reaction imbalances remain in the model despite the significant improvements. A more detailed comparison of the two models, including a comparison of their predictive capabilities, can be found on our GitHub repository: https://github.com/SBRCNottingham/CnecatorGSM/blob/main/JupyterNotebooks/Comparisons_RehMBEL1391.ipynb.

## Metabolic pathways in *i*CN1361

We identified the number of functional metabolic reactions in the model by calculating the minimum ($v_i^{\min}$) and maximum ($v_i^{\max}$) flux values for each reaction using flux variability analysis (FVA), whilst the lower and upper bounds of all transport reactions were left unconstrained. Reactions for which the minimum and maximum flux values are both equal to zero are blocked from carrying flux in *i*CN1361. The total number of active reactions (*i.e.*, $v_i^{\min} < 0$ and/or $v_i^{\max} > 0$) in *i*CN1361 is 850, whilst the remaining reactions are unable to carry flux. The active reactions can be associated to 239 different biosynthesis subsystems in BioCyc. These subsystems include biosynthesis pathways for amino acid metabolism, pyrimidine and purine metabolism, lipid metabolism, cofactors and vitamins metabolism and PHB metabolism. Additionally, the model includes the Entner-Doudoroff (ED) pathway for metabolising sugars into pyruvate and glyceraldehyde-3-phosphate (GAP). Notably, *C. necator* H16 lacks a gene encoding phosphofructokinase (EC 2.7.1.11) and is therefore unable to utilise sugars *via*

the glycolytic Embden-Meyerhof-Parnas (EMP) pathway. Furthermore, the non-oxidative branch of the pentose-phosphate (PP) pathway is also included, whereas the oxidative branch is incomplete because the bacterium lacks a gene coding for phosphogluconate dehydrogenase (EC 1.1.1.44). The model also includes the complete Calvin-Benson-Bassham (CBB) cycle for $CO_2$ fixation that enables autotrophic growth.

In the model, ATP is generated strictly *via* a respiratory electron transport chain (ETC), which agrees with the literature [4,14,35]. The ETC of *C. necator* H16 is highly flexible, allowing for ATP generation under heterotrophic and autotrophic conditions, as well as under both aerobic and anaerobic conditions. The model therefore includes several dehydrogenases that transfer electrons to quinones in the first step of the ETC. Under heterotrophic conditions for instance, either NADH or succinate, both of which are generated from the tricarboxylic acid (TCA) cycle, can be used as electron donors into the ETC, whilst hydrogen or formate donate electrons under autotrophic conditions. The electrons can then be transferred from the quinones to the final electron acceptor *via* several terminal oxidoreductases. During aerobic growth, two quinol oxidases are present (*bo3*-type oxidase and *bd*-type oxidase) that use oxygen as the final electron acceptor. Additionally, ubiquinol-cytochrome-c reductase (CytC) transfers electrons from ubiquinol to cytochrome c. A cytochrome-c oxidase (aa3/cbb3) then relocates the electrons from cytochrome to oxygen. Under anaerobic conditions, the model includes the complete denitrification pathway, which allows for nitrate or nitrite to act as alternative electron acceptors.

A total of 442 reactions are currently blocked in the model. Notably, we found that 130 (29%) of these reactions could be associated to BioCyc degradation pathways, and thus the inclusion of additional transport reactions, and/or further curation for growth on additional carbon sources (beyond this study) may allow feasible flux through many of these reactions. A further 95 (22%) blocked reactions are linked to biosynthesis pathways and thus suggest areas where enzyme-reaction annotations are inaccurate or incomplete (see Table A in S1 Data). Additionally, we found that a total of 130 (29%) of the dead reactions were linked to enzymes with promiscuous activity, such that a functional reaction was present in the model with the same EC number. It is possible that these reactions are correctly predicted as non-functional and may become activated when simulating versions of the model incorporating heterologous genes.

## Validation of *i*CN1361 against experimental data demonstrates high predictive performance for heterotrophic growth conditions

To assess the performance of *i*CN1361 at predicting metabolic phenotypes, we have compared the results from the model against experimental data.

## *i*CN1361 accurately predicts metabolic growth phenotypes for a variety of carbon sources

Firstly, using FBA, we tested the ability of *i*CN1361 to predict the growth phenotype (*i.e.*, growth or no growth) of 131 different carbon sources with known phenotype *in vivo*, as was carried out for the previous GSM of *C. necator* H16 (RehMBEL1391) in [9]. *i*CN1361 predicted the correct phenotype for 62 out of 64 growth supporting substrates and 52 out of 67 non-growth supporting substrates (see S2 Data), which resulted in an 87% overall accuracy. The number of false positives (*i.e.*, the 14 carbon sources supporting growth in the GSM but not *in vivo*) may be due to *C. necator* lacking genes required for transporting these substrates, which, in our analysis, were assumed present to carry out the test.

### *i*CN1361 accurately predicts central carbon metabolism fluxes for growth on fructose

Next, we tested the GSMs ability to quantitatively predict growth rates and intracellular fluxes of *C. necator* H16 growing on fructose. First, we used the experimental data reported in [9] to validate whether the model is capable of predicting growth rates that are comparable to those found *in vivo*. For this simulation, the uptake rate of fructose was constrained using the experimentally determined value (see Fig 1A), whilst all other minimal media nutrients were freely available. As shown in Fig 1A, the GSM predicts growth and oxygen uptake rates that are in-line with the *in vivo* data.

To test whether the optimal solution using FBA involved the correct pathways in the GSM, we compared our results to previously published metabolic flux data obtained using $^{13}$C-labelled fructose [36]. There, the authors employed $^{13}$C-Metabolic Flux Analysis ($^{13}$C-MFA) to quantify the intracellular fluxes of central carbon metabolism for growth on fructose, which provides an independent experimental method for comparing the flux solution predicted using the GSM. First, we aligned 33 reactions from the $^{13}$C-MFA model to the GSM, consisting of the ED pathway, the PP pathway, EMP pathway, the CBB cycle and the TCA cycle. Note that this excludes the anaplerotic reactions (R23, R24, R25, R27, R28 and R29 of Fig 1B), since $^{13}$C-MFA lacks the ability to accurately predict flux through these reactions [37]. To predict the fluxes in *i*CN1361, we then applied parsimonious FBA (pFBA), which identifies the optimal solution in the GSM that maximises growth rate, whilst also minimising the total sum of fluxes.

Importantly, we found a high agreement between the fluxes, and their directionality, for 25 of the 33 reactions (Fig 1B), resulting in a high correlation between the two model predictions ($R^2$ = 0.88, calculated using the Pearson correlation coefficient). The small discrepancy involved reactions in the PP pathway and the CBB cycle. Specifically, the small amount of flux through the PP pathway active in the GSM solution was required for producing 5-phospho-α-D-ribose 1-diphosphate (PRPP), an essential precursor for nucleotides biosynthesis, which is not accounted for in the $^{13}$C-MFA model. The $^{13}$C-MFA, on the other hand, predicted a small amount of flux through the CBB cycle, but this was inactive in the optimal pFBA solution. These differences in flux, however, accounted for only 6% or less of the total fructose flux entering the system in either model. Notably, malate dehydrogenase (Reaction 37 in Fig 1B) contributed the largest discrepancy, carrying 33% of the total fructose flux in the pFBA solution but no flux in the $^{13}$C-MFA solution.

Furthermore, we also tested whether the $^{13}$C-MFA predictions were feasible in the GSM, if considering alternative sub-optimal solutions. To this aim, we carried out flux variability analysis (FVA) to calculate the minimum and maximum flux values for each of the 33 reactions in *i*CN1361, whilst constraining the growth rate to a minimum of 90% of the optimum value (Fig 1C). Importantly, all $^{13}$C-MFA determined fluxes were reachable in *i*CN1361 (including malate dehydrogenase that was responsible for the largest disagreement between the two models), with the exception of 3 reactions of the TCA cycle (citrate synthase, fumarate hydratase and aconitate hydratase), 2 reactions from the PP pathway (transketolase and ribulose-phosphate 3-epimerase) and 2 reactions involved in glycolysis (phosphoglycerate kinase and glyceraldehyde-3-phosphate dehydrogenase). The reason for these discrepancies is possibly due to the $^{13}$C-MFA being limited to central carbon metabolism and thus excluding other important processes, whilst also neglecting cofactor balancing. The results from the pFBA and FVA simulations are provided in S3 Data.

Finally, we also analysed the fluxes through the ETC that were active in the FBA optimal growth solution. Experimental flux values are not available for these reactions; however, we

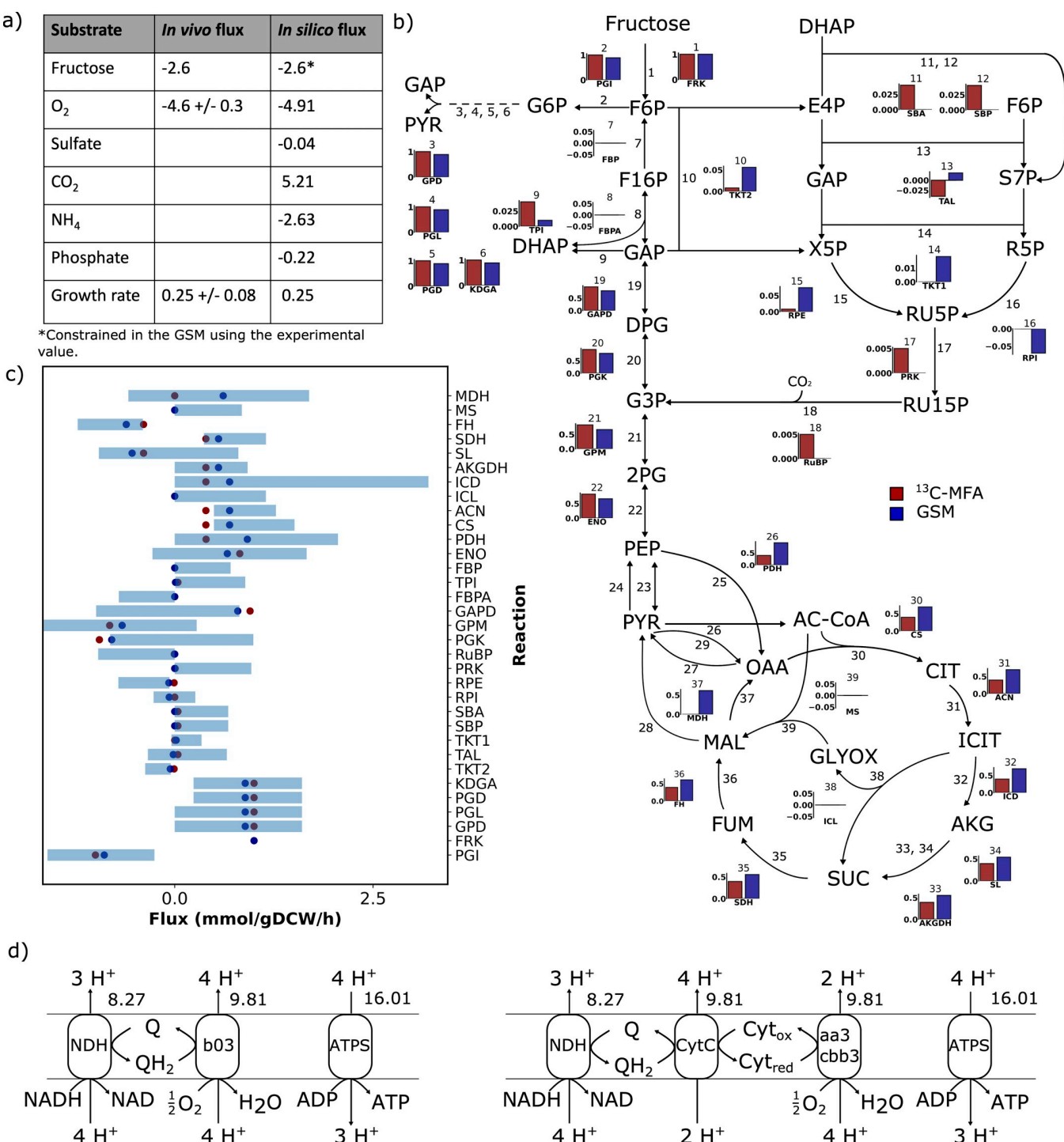

**Fig 1. Experimental validation of central carbon metabolic fluxes for the *i*CN1361 GSM for growth on fructose.** a) GSM prediction of substrate uptake rates and growth rate using *i*CN1361 for growth data provided in [9]. Units for the substrates are in mmol/gDCW/h, whereas the growth rate is h$^{-1}$. Note that the uptake rate of fructose was constrained in the model using the experimentally determined value. b) Comparison of the fluxes predicted by *i*CN1361 (blue bars) and $^{13}$C metabolic flux analysis (red bars) for growth on fructose mineral media. The flux in the GSM have been normalised to correspond to a fructose uptake of 1 mmol/gDCW/h to allow comparison with the $^{13}$C-MFA data. c) Minimum and maximum flux values for each reaction, calculated using flux variability analysis. Blue data points correspond to the pFBA flux prediction, whereas red dots correspond to the $^{13}$C-MFA flux prediction. d) Two feasible routes through the electron transport chain of *C. necator* H16 predicted using *i*CN1361 growing at the maximum growth rate. The flux units are all in mmol/ gDCW/h. See the abbreviations section for the metabolite and reaction names.

can estimate the amount of ATP produced per oxygen atom consumed (P/O ratio). Two alternative routes were found available for oxidative phosphorylation for optimal growth on fructose (Fig 1D), both of which result in a P/O ratio of 1.63. This is close to the P/O ratio determined for aerobic growth of *Escherichia coli*, which was estimated at 1.5 [38]. The route involving NADH dehydrogenase (NDH) and ubiquinol oxidase (b03), is analogous to the route in *E. coli*, which couples a proton motive force to the transfer of electrons from NADH to oxygen. The alternative route transfers electrons from NADH to oxygen, whilst utilising NADH dehydrogenase, ubiquinol-cytochrome-*c* reductase (CytC) and cyto-chrome-*c* oxidase (aa3/cbb3). Interestingly, the latter route has previously been found active during conditions of high oxygen availability [39], and thus *i*CN1361 can be constrained accordingly.

## TraDIS analysis provides a genome-scale *in vivo* assessment of *i*CN1361

To test the ability of *i*CN1361 at determining gene KO phenotypes, we have performed TraDIS [29], which is a high-throughput *in vivo* approach for determining gene essentiality in bacteria. To this end, a library of over 1 million *C. necator* H16 transposon mutants was constructed and the number of insertions per kilobase per million (IPKMc) was calculated for each CDS, as previously described by Hwang et al. (2018) [40]. A $\log_2$(IPKMc+1) (insertion index) threshold of 3.9 was determined as described in the Methods section, so that the *C. necator* H16 genes with insertion indexes equal or lower than this cut-off value ($\log_2$(IPKMc+1) $\leq$ 3.9) were identified as essential (see Table A in S4 Data and Methods for details). Similarly, a $\log_2$(IPKMc+1) threshold of 4.4 was determined, such that *C. necator* H16 genes with an insertion index greater or equal to this cut-off ($\log_2$(IPKMc+1) $\geq$ 4.4) were classified as non-essential. Genes with an insertion index falling in between the 3.9 and 4.4 thresholds were not classified as essential or non-essential. Also note that, due to low sequencing coverage of some genomic regions of *C. necator* H16, a total of 204 and 198 genes were excluded from the TraDIS analysis for the super optimal broth (SOB) and fructose mineral media (FMM) conditions, respectively (see Table A in S4 Data for details). Notably, however, only 20 and 15 of these genes are involved in the GSM, respectively.

Gene essentiality estimations have recently been published for *C. necator* H16 grown on fructose, as well as on other carbon sources [24], using Tn-Seq analysis. Notably, however, our TraDIS approach uses a significantly larger transposon mutant library (over 1 M mutants compared to approximately 60,000 mutants in the Tn-Seq analysis–see Table 2 for a detailed comparison of these two datasets), and thus provides a more complete coverage of the *C. necator* H16 genome. Nevertheless, we have used the data to further validate our gene classifications, particularly for cases where the GSM and TraDIS analysis differed, as discussed in the next section.

For clarity, we herein refer to an experimental gene deletion as a gene knockout and a computational gene deletion as a gene deactivation.

**Table 2. Comparison of transposon mutant library sizes and average insertion frequencies between TraDIS and the Tn-Seq analysis from [24].**

| Library | Size of coding regions (bp) | Number of genes analysed | Average gene length (bp) | Total insertions | Insertions after curation | Unique insertions | Average insertions per gene |
|---|---|---|---|---|---|---|---|
| TraDIS (FMM) | 6,454,517 | 6,637 | 972.2 | 9,415,576 | 7,158,925 | 1,335,710 | 201.2 |
| TraDIS (SOB) | 6,454,517 | 6,637 | 972.2 | 8,315,651 | 6,226,973 | 1,050,800 | 158.2 |
| Tn-Seq (LB) | 6,106,532 | 6,549 | 932.4 | 107,708 | 84,761 | 60,000 | 9.2 |

### *i*CN1361 accurately predicts gene essentiality under heterotrophic growth

Using the results from the TraDIS analysis, we have experimentally validated the gene essentiality predictions of *i*CN1361. Here, we predicted *in silico* the set of genes, amongst those included in the model, that are essential for growth on FMM.

In Table 3 we show the performance of the FBA predictions for gene essentiality, as compared to TraDIS results, in terms of the number of true (T) and false (F) predictions for essential (P) or non-essential (N) phenotypes. The overall accuracy of 92%, calculated from the confusion matrix, provides evidence that the model is capable of predicting gene knockout phenotypes with comparable accuracy to the well-curated *E. coli* model [41].

Using the overall accuracy as a measure of performance, however, can lead to bias when considering unbalanced classes. Therefore, precision and recall performance measures were also considered. The precision measure shows that 81% of the genes predicted to be essential in the model were also identified as essential in TraDIS. On the other hand, recall shows that only 62% of the TraDIS essential genes were also required for growth on fructose in the model. For the sake of comparison, we also tested the accuracy of *i*CN1361 using the Tn-Seq data and found similar results for the overall accuracy (89%) and precision (77%) scores but a weaker recall score (51%). The relatively low performance *via* recall in the model compared to both approaches is due to the number of genes being classified as non-essential in the model but identified as essential using the TraDIS-based approaches. This underestimation of the number of essential genes predicted by *i*CN1361, with respect to the TraDIS *in vivo* analysis, was not entirely unexpected. Indeed, the model does not consider information about gene expression regulation and, as a result, alternative pathways or isoenzymes identified in the GSM may not carry enough flux to sustain *C. necator* H16 growth *in vivo*. The number of false-negative genes coding for isoenzymes that are potentially inactive or poorly expressed during growth on fructose *in vivo* was therefore further investigated. Notably, using the GSM we found that 30 of the 79 FN genes were associated with at least 1 essential reaction that could be catalysed by at least 1 other isoenzyme. Using previously published RNA-Seq data obtained for *C. necator* H16 growing on fructose [42], we compared the expression levels of genes encoding for isoenzymes associated with reactions identified as essential in the GSM. For each of these reactions, the most expressed isoenzyme-encoding gene was identified, and any other isoenzyme-encoding gene, which showed expression levels at least 5-fold lower than the most expressed gene (see Table D in S4 Data), was subsequently deactivated in the model. The constrained model was then used to update the GSM gene essentiality predictions, and, importantly, showed that gene expression integration was able to improve the recall performance metric to 69%. The isoenzymes that were associated to essential reactions but had low gene expression and thus deactivated in the model, may indicate inaccurate enzyme-reaction annotations. Alternatively, however, the isoenzymes may each have unique kinetic properties that are required for varying the metabolic flux activity and/or direction, and thus may be important for growth under different environmental conditions or during specific growth phases. Even though additional information is required to further investigate the role of isoenzymes, constraining the model using gene expression data for specific growth conditions currently offers a valuable solution, as demonstrated here.

**Table 3. Confusion matrix showing the number of true positives (TP), true negatives (TN), false positives (FP) and false negatives (FN), obtained by comparing TraDIS results with GSM predictions.**

|                                        | Essential in GSM | Non-essential in GSM |
|----------------------------------------|:----------------:|:--------------------:|
| Essential in TraDIS experiments        | 127 (TP)         | 79 (FN)              |
| Non-essential in TraDIS experiments    | 29 (FP)          | 1080 (TN)            |

We compared the false-negatives to the Tn-Seq analysis and found that 55 of the 79 FN genes were also essential in the Tn-Seq data. One of the intrinsic limitations of TraDIS-related approaches (including Tn-Seq), however, is that transposon insertions in genes associated with viable but slow-growing gene knockout mutants are often under-represented in the sequencing results, since these mutants are outcompeted by faster growing strains. As a result, these genes may be misclassified as essential in the TraDIS predictions [29]. Therefore, we tested whether any FN genes were predicted to be growth-limiting in the model (*i.e.*, reduced the growth rate in the GSM to 85% of the wild-type growth rate or lower). The GSM predictions show that deactivating any of the 9 genes encoding the NADH dehydrogenase complex, or any of the 4 genes coding for the succinate dehydrogenase subunits, results in inactive enzymes, which severely affect the *in silico* growth rate. Indeed, even though alternative solutions exist in the model, these are not as efficient at transferring electrons as the NADH and succinate dehydrogenases, which play key roles in the ETC. For example, the model predicts that additional membrane-bound enzymes, including NADH type-II, alcohol, glycerol and proline dehydrogenases, are able to partially restore the flux through the ETC *via* electron donation from NADH, ethanol, glycerol and proline, respectively. The availability of these substrates, particularly the amino acid proline, however, may be tightly regulated with amino acid biosynthesis being required for growth and thus may not be feasible *in vivo*.

Additionally, the GSM predicted that deactivation of the 2 genes encoding subunits of the 2-oxoglutarate dehydrogenase complex, which is part of the TCA cycle, does not result in an attenuated growth phenotype, but increases the demand for oxygen. The alternative solution predicted by the model involves the activation of the glyoxylate shunt, which enables the replenishment of the essential TCA metabolites (succinate and malate). Additionally, the GSM also predicted increased oxygen uptake when deactivating the *H16_A1188* gene, coding for a 2-phospho-D-glycerate hydrolase, which catalyses the conversion of 2-phosphoglycerate (2-PG) to phosphoenolpyruvate (PEP) in one of the final steps of the ED pathway, leading to pyruvate production. An alternative route in the model exists, however, that utilises the serine biosynthesis pathway to convert 3-phosphoglycerate (G3P) to serine, and then subsequently converts serine to pyruvate *via* serine ammonia-lyase, which thus bypasses the last two steps of the ED pathway. However, as discussed for proline, the flux through the biosynthetic pathway for the amino acid serine is again likely controlled by tight regulation. Therefore, this alternative route may not be feasible *in vivo*.

Further inspection into the 33 remaining FN genes (see Table D in S4 Data), revealed several cases where again missing regulatory information and/or kinetic parameters are likely limiting the GSM predictability. Other FN genes were involved in the oxidative stress response. Since the setup of the FBA formulation only considers biomass components as essential cellular processes, incorrect classification of genes involved in stress response is to be expected. Some of the FN genes may indicate areas of the model that are still incomplete and may require additional curation. Several FN genes, for instance, were involved in dTDP-rhamnose biosynthesis, which is a nucleotide sugar used in the O-antigen component of lipopolysaccharides (LPS). The LPS composition in *C. necator* H16, however, is not well defined and the LPS biosynthesis pathway is poorly annotated. For that reason, a generic pathway from *E. coli* was used in *i*CN1361. These forementioned discrepancies, thus point to areas of future development of the model, which will become possible as gene annotation improves and as data availability increases to enable the integration of the GSM with more advanced modelling techniques that can account for enzyme kinetics and regulation.

Despite these discrepancies, the *i*CN1361 achieves a high overall accuracy for predicting gene essentiality phenotypes that is comparable to the performance of the well-curated *E. coli* model, *i*AF1260 [41].

## Experimental verification of *C. necator* null mutants for selected FP genes

Essentiality predictions produced by *i*CN1361 disagreed with *in vivo* observations also for a set of 29 FP genes (see Table C in S4 Data), which were not identified as essential for growth in FMM using TraDIS. Out of these 29 false positive predictions, 19 were also classified as false positives when comparing to the Tn-Seq data, whilst 9 were predicted as essential in agreement with the GSM predictions. To verify which of these approaches was correct, 6 of these genes were inactivated in *C. necator* H16. The insertion index values observed following growth in FMM for 3 of these genes (*H16_A0792*, *H16_A3408* and *H16_A3434*) were just above the non-essentiality classification threshold ($\log_2$(IPKMc+1) > 4.4), since they ranged between 4.64 and 5.13, while the remaining 3 genes (*H16_A3038*, *H16_A3084* and *H16_A3165*) showed very high insertion index values ($\log_2$(IPKMc+1) > 10). The *H16_A0792* (*pheA*) and *H16_A3434* (*aroB*) genes code for a prephenate dehydratase and a 3-dehydroquinate synthase, respectively, and are involved in the biosynthesis of the aromatic amino acids tryptophan, phenylalanine and tyrosine. In particular, *aroB* is involved in the synthesis of all three amino acids, while *pheA* only plays a role in the production of phenylalanine and tyrosine. The other gene characterised by an insertion index of around 5, *H16_A3408* (*hisE*), codes for a phosphoribosyl-ATP pyrophosphatase and is also involved in amino acid metabolism (histidine biosynthesis). The remaining genes, *H16_A3038* (*nadA*), *H16_A3084* (*panB*) and *H16_A3165* (*ubiC*) are linked to different cellular processes (biosynthesis of NAD/NADP; (R)-pantothenate and CoA; and ubiquinone) and respectively encode a quinolinate synthetase A protein, a ketopantoate hydroxymethyltransferase and a chorismate lyase. To test whether any of these enzymes was essential for growth, *C. necator* single mutants in each of these genes and their parental strain *C. necator* H16 wild type were cultivated in SOB and FMM (Fig 2).

As shown in Fig 2, all 6 mutants retained the ability to grow in SOB complex medium and showed growth rates comparable to the wild-type strain. These results are consistent with the TraDIS data since the insertion index values calculated for these genes under these growth conditions (ranging between 9.76 and 11.77) were all significantly higher than the essentiality threshold. On the other hand, none of the mutants managed to grow in FMM. This outcome was somehow expected for the *ΔA0792*, *ΔA3408* and *ΔA3434* mutants since the FMM insertion indexes associated with the genes inactivated in these strains were very close to the essentiality cut-off and most of the transposon insertions identified in these genes mapped in proximity of their 5' and 3' ends. However, the *nadA*, *panB* and *ubiC* genes showed FMM insertion indexes over 2-fold higher than the essentiality threshold, with values comparable to those observed following growth in SOB. Therefore, according to the TraDIS predictions, the *nadA*, *panB* and *ubiC* genes are not essential and the *ΔA3038*, *ΔA3084* and *ΔA3165* mutants should have been able to grow in FMM. Notably, 2 out of these 3 genes (*H16_A3038* and *H16_A3165*) were also incorrectly predicted as non-essential in the Tn-Seq data. These incongruencies could be explained as the result of an intrinsic limitation of the TraDIS approach. Indeed, in these experiments, a large number of different strains are pooled together under growth conditions of interest, where they are made to compete for limited amounts of resources. Within this bacterial population there would be a subset of cells that is unfit for this specific ecological niche and would therefore die and, eventually, lyse. The metabolites released in the culture medium by lysed cells or excreted by viable mutants that may accumulate intermediates as a consequence of metabolic pathways disruption, become available to the rest of the bacterial population. It is possible that, in our TraDIS experiment, traces of nicotinate, 2-dehydropantoate/(R)-pantoate and 4-hydroxybenzoate may have been present in FMM at sufficiently high concentrations to sustain the growth of null mutants for the *nadA*, *panB* and *ubiC* genes, respectively, thus by-passing the need for the essential enzymes encoded by these

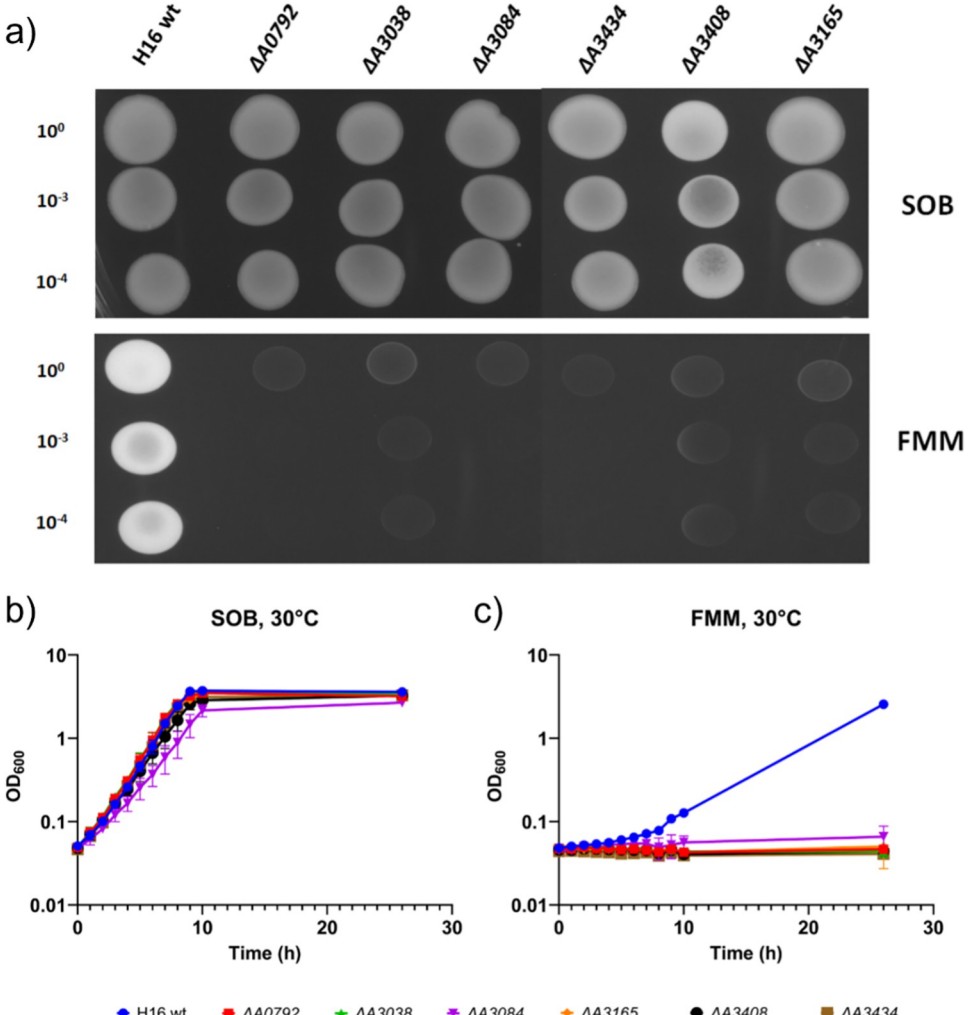

**Fig 2.** Growth of *C. necator* H16 wild type (H16 wt) and its isogenic mutants *ΔA0792*, *ΔA3038*, *ΔA3084*, *ΔA3165*, *ΔA3408* and *ΔA3434* on a) SOB (top panel) and FMM (bottom panel) plates, and liquid SOB (b) and FMM (c). Details about cultures set up are reported in the Methods section.

genes. However, when cultures were set up in FMM starting from clonal populations of the *ΔA3038*, *ΔA3084* and *ΔA3165* mutants, these failed to grow since they were unable to synthesise the essential metabolites NAD/NADP, (R)-pantothenate/CoA and ubiquinone, respectively. Interestingly, most of the other genes involved in these three metabolic pathways, especially those controlling the enzymatic reactions located downstream of the steps catalysed by the products of the *nadA*, *panB* and *ubiC* genes, were identified as being essential for growth in both SOB and FMM by the TraDIS approach. These observations suggest that availability of other key intermediates of the NAD/NADP, (R)-pantothenate/CoA and ubiquinone biosynthetic routes (*e.g.*, deamino-NAD$^+$, (R)-pantothenate and 3-polyprenyl-4hydroxybenzoate, respectively) in FMM during the course of the TraDIS experiment, may not have been sufficient to rescue the *C. necator* mutants unable to produce these metabolites. Alternatively, the uptake rates for these metabolic intermediates may have been significantly slower than their biosynthetic rates, under these experimental conditions. As a result, mutants for the genes controlling the synthesis of deamino-NAD$^+$, (R)-pantothenate and 3-polyprenyl-4hydroxybenzoate (*H16_A0913/nadD*, *H16_A2959/panC* and *H16_A3107/ubiA*, respectively) were

outcompeted by faster growing strains. Consequently, only very few transposon insertions could be detected in these genes at the end of the TraDIS experiment. This limitation may be responsible for the misclassification of genes as non-essential in both our TraDIS and the Tn-Seq approaches. The small number of discrepancies found between the two transposon-sequencing based approaches, however, is likely the result of using slightly different experimental conditions.

Overall, these results indicate that GSM and TraDIS offer complementary approaches for the identification of essential genes in bacteria and should therefore be used in combination to obtain more reliable gene essentiality predictions.

## Integration of transcriptomics data to investigate nitrogen-limited conditions

To demonstrate the relevance of the curated GSM for biotechnological applications, we employed *i*CN1361 to investigate the metabolic changes occurring in *C. necato*r H16 during nitrogen-limited conditions, where carbon flux is re-directed towards the PHB biosynthesis pathway. This pathway consists of 3 enzymes, β-ketothiolase (encoded by *phaA*), acetoacetyl-CoA reductase (encoded by *phaB*) and PHB synthase (encoded by *phaC*), which convert acetyl-CoA to PHB, whilst consuming NADPH, and thus provides a sink for storing the excess carbon and energy generated from the Entner-Doudoroff pathway.

Applying standard FBA however, whilst mimicking nitrogen-limited conditions, resulted in pyruvate and lactate production rather than PHB. Notably, pyruvate production is in-line with experimentally observed behaviour of PHB-negative strains [43]. By applying FVA, we found that several alternative solutions are available in the model under nitrogen-limited conditions, which can result in various by-products, including acetate, lactate, pyruvate, succinate, formate, ethanol, hydrogen, propionate and also PHB (Fig 3). Therefore, we next applied flux sampling analysis to *i*CN1361, which generates a probability distribution for each reaction and thus allows us to explore the likeliness that each product is synthesised [44]. Using this approach, however, we found that the probability distribution of 505 reactions were significantly different across repeated sampling runs, which suggests a lack of convergence due to the high number of alternative solutions. To reduce the number of alternative solutions, and hence the variety of different by-products, we therefore integrated RNA-Seq data derived from nitrogen-depleted conditions to generate a condition-specific version of *i*CN1361, which is highly constrained according to the expression levels of the corresponding genes. We will refer to *i*CN1361, without RNA-Seq integration, as the base model from this point forward to distinguish it from the condition-specific models.

## Condition-specific models for growth- and PHB-phases

In this work, we integrated the transcriptomic data reported in [42] with *i*CN1361 to generate growth phase and PHB-phase specific models by employing the well-established tool iMAT [45,46]. This approach searches for a solution that maximises the agreement between the active fluxes and the transcription data. Specifically, metabolic reactions are categorised as lowly, moderately or highly active based on the expression levels of their encoding genes.

To run iMAT, we first downloaded the reads per kilobase per million (RPKM) values of the genes for both growth phase (f16) and nitrogen-limited (or PHB-producing) phase (f26) from https://www.ncbi.nlm.nih.gov/geo/query/acc.cgi?acc=GSE47759. Lower bound thresholds of 212 and 21 were determined for classifying reactions as lowly expressed, whilst upper bound thresholds of 11439 and 17535 were determined for classifying reactions as highly expressed, for the f16 and f26 conditions, respectively (see Methods for details on threshold

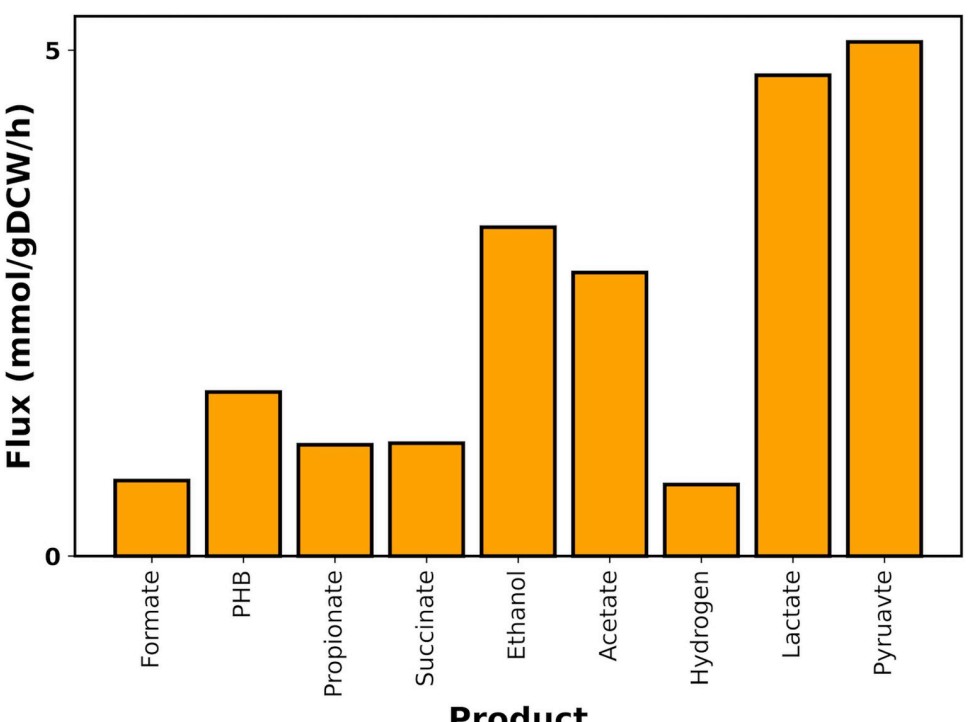

**Fig 3. The predicted range of fluxes for by-products in *i*CN1361 using flux variability analysis, whilst simulating nitrogen-limited conditions.**

determination). The reactions with expression values that lie in between the low and high threshold were then categorised as moderate. Growth rates of 0.2 h$^{-1}$ and 0.009 h$^{-1}$ were estimated for the f16- and f26-phase conditions, respectively, using the residual biomass curve in [42] (see Methods for details), which were then used to constrain the model. The minimum amount of nitrogen required to achieve a growth rate of 0.009 h$^{-1}$ was also used to constrain the model for generating the f26-condition specific GSM. The fructose uptake rate was fixed at 2.1 mmol/gDCW/h, which corresponds to the minimum fructose required for achieving a growth rate of 0.2 h$^{-1}$. Note that the fructose uptake rate for the nitrogen-limited condition was assumed to remain at the same level as the growth condition. All transport reactions were unconstrained in the direction of production to test whether the approach was capable of correctly predicting PHB as the main by-product. iMAT was then applied to the model constrained according to each condition, resulting in two condition-specific models (hereafter named *i*CN1361-f16 (growth phase) and *i*CN1361-f26 (nitrogen-limited phase), which are provided on https://github.com/SBRCNottingham/CnecatorGSM/tree/main/JupyterNotebooks/Data).

## Predicting the metabolic changes during nitrogen-limited conditions

Next, using the two condition-specific models we investigated the differences in metabolic fluxes between the two conditions. For this analysis, we again applied flux sampling to each model to explore the possible range of feasible flux values for each reaction. Unlike the sampling results of the *i*CN1361 base model, only 3 reactions in the iMAT models were not able to converge in the sampling results. The probability distribution for each reaction was subsequently used to assess whether the flux had significantly altered between the two conditions.

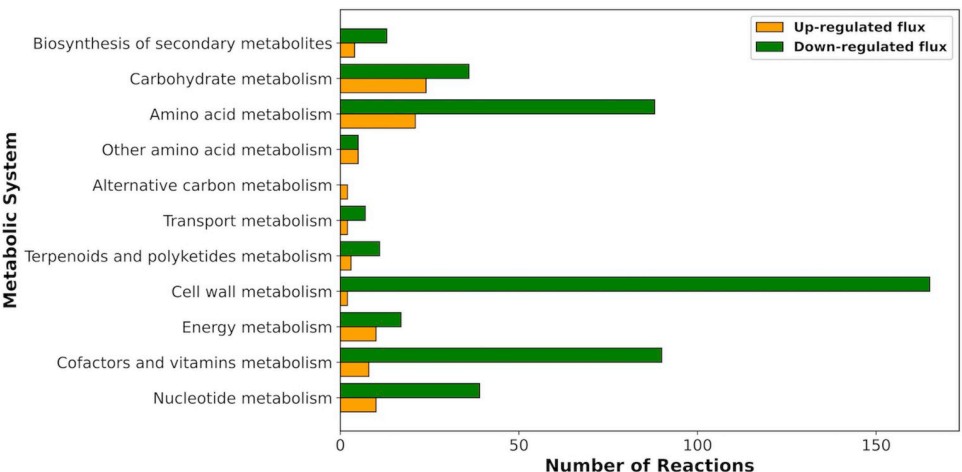

**Fig 4. The number of reactions predicted to be up-regulated (orange) or down-regulated (green) in the nitrogen-limited conditions using the genome-scale metabolic model analysis.**

To select the reactions with differentially altered flux, we used the approach outlined in [47], which uses the Kolmogorov-Smirnov (KS) test to compare flux distributions. Reactions that are significantly different, which also have a high fold-change (*i.e.*, the ratio of the mean flux value of the samples for each condition) were selected for further analysis. The reactions with a negative fold-change are those that are predicted to have a down-regulated flux, whereas those with a positive fold-change are predicted to have an up-regulated flux.

Using this approach, we found 66 reactions that were significantly up-regulated in the *i*CN1361-f26 (nitrogen-limited) model, and 396 reactions that were significantly down-regulated. A large proportion of the down-regulated reactions involved reactions associated with biomass-related processes, which are essential for growth (as shown in Fig 4). A total of 390 of these reactions, in fact, showed minimal variation in the flux values across all samples for each condition (standard deviation < 0.001), suggesting that reaction activity is regulated with the growth state of the cell.

Importantly, included in the up-regulated reactions was the poly-hydroxybutyrate (PHB) biosynthesis pathway. The method correctly predicted that excess carbon in the nitrogen-limited *i*CN1361-f26 model is redirected towards PHB biosynthesis (ranging between 1.89 and 2.2 mmol/gDCW/h). In addition, no other by-products were produced in the GSM simulations, which is in agreement with previously reported *in vivo* observations [48]. Additionally, the model predicted that the increased production of acetyl-CoA, which is the precursor to PHB (Fig 5), is achieved *via* a decrease in flux through the TCA cycle, in agreement with [49], as well as an increased flux through pyruvate dehydrogenase. Notably, however, despite the significant decrease in fluxes, the TCA cycle is still active, in agreement with [48].

Furthermore, the reactions of the CBB cycle, which are involved in $CO_2$-fixation, were also up-regulated in the *i*CN1361-f26 model, in agreement with the experimental results in [42,48]. The reason why the CBB cycle is activated during heterotrophic growth conditions is unclear, therefore we investigated whether the activation of the CBB cycle could be associated to other metabolic pathways. Here, we used the Pearson's correlation coefficient to identify reactions whose probability distribution from the flux sampling was highly correlated to the probability distribution of the ribulose biphosphate carboxylase (RuBisCo). From these results, we found an almost perfect anti-correlation to the reactions of the ED pathway (r < -0.99), which is due to the flux from glucose-6-phosphate isomerase being redirected towards the pentose

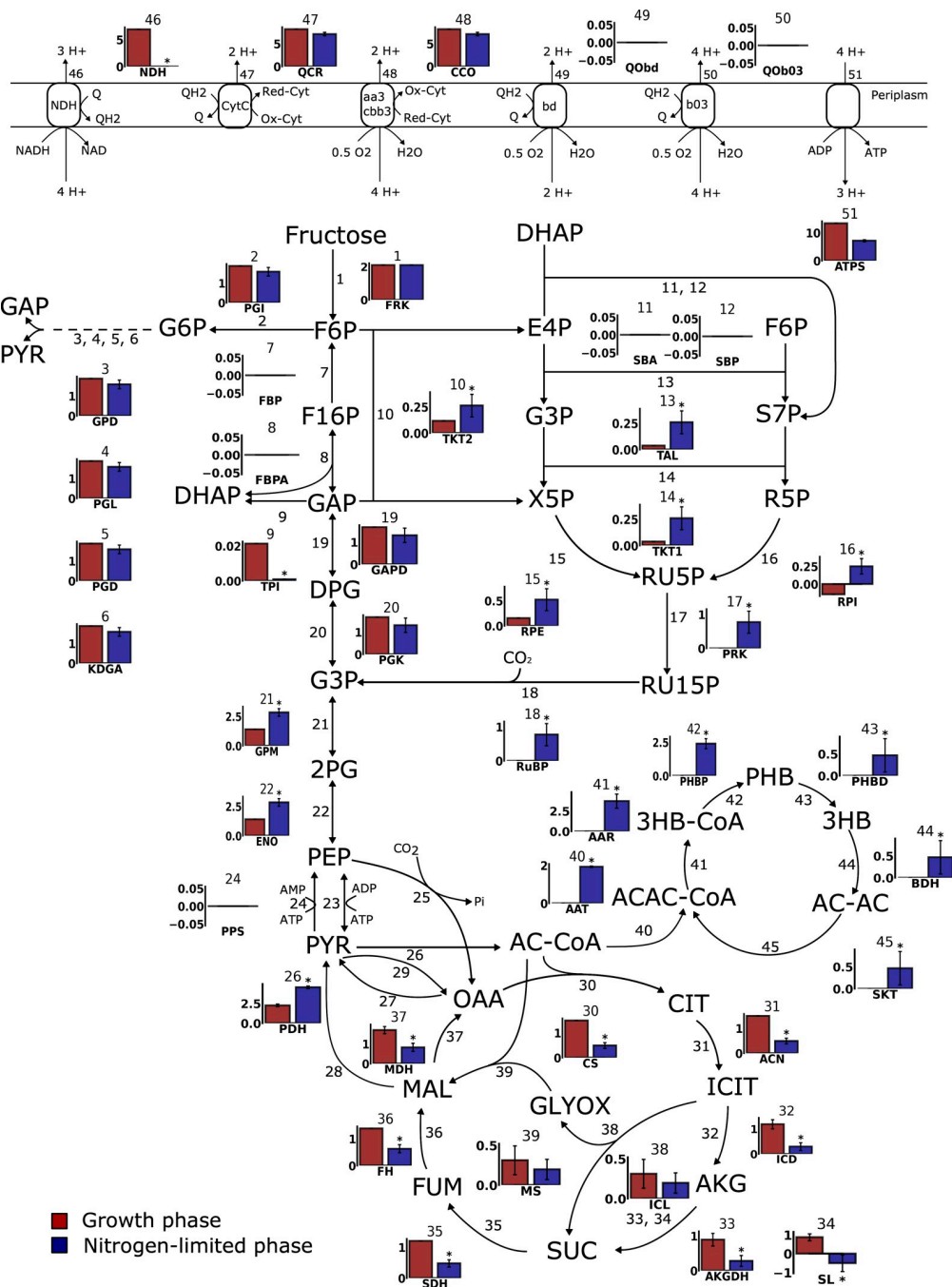

**Fig 5. Flux diagram of the central carbon metabolism, electron transport chain and the PHB cycle comparing the predicted flux in the growth and nitrogen-limited phase.** The bar charts show the mean flux value from the flux sampling simulated on *i*CN1361-f16 (red) and *i*CN1361-f26 (blue). Reactions that were predicted to have significantly up-regulated or down-regulated flux in *i*CN1361-f26 are highlighted with an asterisk. See the abbreviations section for the metabolite and reaction names.

phosphate pathway (Fig 5). More interestingly, however, was the high correlation found to reactions in the TCA cycle (r > 0.76) and pyruvate dehydrogenase (r > 0.8), which supports the idea that the bacteria activate $CO_2$-reutilising reactions to maximise PHB production [50]. Notably, the probability distributions of the CBB reactions were not highly correlated to the

PHB pathway reactions. However, by deactivating the RuBisCo reaction from *i*CN1361, whilst maximising for PHB production, we could show that the PHB yield was reduced by 16%, similar to the experimental results from [50].

Furthermore, quite unexpectedly, the fluxes through the reactions involved in PHB-degradation (Reactions 43–45 in Fig 5) were also up-regulated in *i*CN1361-f26. In this pathway, the PHB is recycled back to acetoacetate by 3-hydroxybutyrate (3-HB) dehydrogenase, whilst also releasing an NADH (Reaction 44 in Fig 5). Succinyl-CoA transferase is then utilised to convert acetoacetate to acetoacetyl-CoA, whilst also converting succinyl-CoA to succinate (Reaction 45 in Fig 5). The cycling of flux through the PHB biosynthesis and PHB degradation pathways may therefore act as a transhydrogenase for balancing the level of NADPH to NADH during nitrogen-limited conditions. The excess NADH can then be consumed by the CBB cycle for sinking the excess electrons, as discussed above. Interestingly, PHB recycling is also supported by the results in [51], which validated experimentally that PHB production and consumption can occur simultaneously.

Other reactions with up-regulated flux levels included those involved in the methylcitrate cycle, which converts propionyl-CoA to pyruvate and succinate. These reactions may become activated during nitrogen-limited conditions to detoxify the cell from toxic levels of propionyl-CoA, which results from the down-regulation of flux through fatty acid biosynthesis pathways. Alternatively, however, the pathway may be activated for replenishing the TCA cycle metabolites.

A number of reactions involved in glutamate, alanine and aspartate metabolism also had up-regulated flux levels. These metabolites are nitrogen-rich compounds and thus changes to their metabolism may be required for scavenging during nitrogen starvation. Similarly, reactions involved in purine metabolism also had up-regulated flux levels, which again may be required for scavenging. Alternatively, however, some of these reactions are also involved in consumption of GTP *via* adenylosuccinate synthase and may therefore be important for regulating the levels of GTP, which has previously been found to be important for survival of bacteria during stress [52].

Finally, the predicted up-regulated fluxes also included reactions of the NAD salvage pathway, which replenishes the essential NAD molecules by recycling degraded NAD products, and thus bypassing the *de novo* NAD biosynthesis pathway from aspartate.

## Predicted regulatory control of metabolic pathways for designing engineering strategies

Understanding the type of control on reactions is important for identifying promising targets for optimising flux towards a product of interest in metabolic engineering applications. Over-expressing a gene, which is predicted to increase flux towards the target product, for example, would be non-effective if its associated reaction is regulated by feed-back inhibition. By comparing the changes in flux to the changes in gene expression between the conditions f16 (growth phase) and f26 (nitrogen-limited phase), we can predict whether a reaction is controlled at the transcriptional or post-translational level, in a similar way to the approach described in [53]. Using this approach, we predicted that 143 (31%) of the up- and down-regulated reactions under nitrogen limitation were likely controlled at the post-translational level.

For validation purposes, we have compared the type of regulatory control predicted for 14 amino acids biosynthetic pathways for which regulation is reported in the EcoCyc database [54] and that are conserved in *C. necator* H16. Importantly, the prediction of transcriptional-level regulation, post-translational-level regulation, or a combination of the two, for the 14 amino acid biosynthesis pathways, shown in Table C in S5 Data, was in high agreement with

the regulation outlined in EcoCyc, suggesting these pathways are controlled in a similar way to *E. coli*.

Next, we analysed the type of regulation predicted for PHB metabolism. First, the data from Shimizu et al. (2013) [42] showed that the genes encoding the PHB biosynthesis pathway, *phaA* and *phaB* (Reactions 40 and 41 in Fig 5), were highly expressed under both the f16 and f26 conditions, whereas up-regulation at the flux level was only predicted for the nitrogen-limited model. This result suggests that PHB biosynthesis is regulated at the post-translational level to avoid PHB accumulation competing with cellular growth, which is in-line with the regulation described in [49] that reported inhibition of β-ketothiolase (Reaction 40 in Fig 5) by CoA. Evolving this enzyme to prevent CoA inhibition, as well as increasing the concentrations of the precursor acetyl-CoA and the cofactor NADPH, may therefore offer possible strategies for increasing PHB (and more generally PHAs) yields in *C. necator* H16. Notably, however, *phaC* was only moderately expressed during the growth phase, and subsequently increased in the nitrogen-limited phase ($\log_2$ fold change = 1.8) suggesting that PHA synthase may also be a limiting factor. Over-expression of the *phaC* gene may thus be a suitable target for increasing PHAs.

The results from the flux analysis also suggested that flux to acetyl-CoA, the precursor to the PHB pathway, was increased *via* pyruvate dehydrogenase in the *i*CN1361-f26 (nitrogen-limited) model. Expression of the genes associated to this reaction, however, is down-regulated in the f26 condition, and thus over-expression of these genes may provide opportunities for increasing PHB *in vivo*. This strategy has already been demonstrated experimentally using an *E. coli* strain expressing the *C. necator* H16 PHB pathway genes [55].

Additionally, the genes involved in increased flux through the CBB cycle in the flux analysis were up-regulated in the f26 (nitrogen-limited phase) gene expression data [42], which suggests transcriptional regulation. Over-expressing these genes during the optimal growth phase may therefore offer an opportunity for reducing the $CO_2$ waste from fructose catabolism, which is then redirected towards PHB or any target product of interest. Increasing product yields by introducing $CO_2$-fixing pathways to heterotrophic organisms has previously been suggested for reducing waste $CO_2$ in [56]. An additional electron source, such as $H_2$, however, is likely required to provide the energy to drive the CBB cycle, as also suggested in [24].

The flux analysis also predicted activity through the PHB degradation pathway, in-line with the up-regulated gene expression in the f26 (nitrogen-limited) condition [42]. The predicted ratio of PHB consumption to production rate, however, was very low, and maintaining this balance likely requires post-translational regulation. Indeed, a previous study suggested that PHB depolymerase activity in *C. necator* H16 is inhibited, either directly or indirectly, by the signalling molecule alarmone (p)ppGpp [57] since increased intracellular levels of this molecule resulted in improved PHB accumulation.

The predicted regulatory control of the flux for all up- and down-regulated reactions are provided in Table B in S5 Data, and, importantly, provide useful information for assessing whether a flux increase through a target reaction would be more readily achieved using interventions at either the transcriptional (*i.e.*, gene over-expression) or post-translational (*i.e.*, enzyme evolution) levels. Such strategies can be applied for increasing PHB, or even more interestingly, PHA yields in *C. necator* H16, through the up-regulation of 3-hydroxybutyrate (3HB) and other hydroxy-alkanoate CoAs.

## Discussion

The natural ability of *C. necator* H16 to grow on a variety of low-cost feedstocks, including $CO_2$, whilst also producing large quantities of PHB during nutrient-limited conditions, make

it a particularly attractive host for producing biochemicals and biofuels [4]. However, identification of genetic modifications and cultivation conditions for its optimal application as a microbial cell factory requires a comprehensive system-level understanding of its metabolism. A curated genome-scale metabolic model is fundamental for such a task. A major challenge in GSM development and analysis, however, is the readability and reusability of models. Often GSMs, including the original GSM of *C. necator* H16 [9], are constructed using non-systematic identifiers for metabolites and reactions, which considerably complicates the application of analysis packages, interpretation of results and further improvements of the model. Although a huge effort was recently made to improve the model readability of RehMBEL1391 [24], the lack of identifiers and chemical formulae for a significant subset of metabolites, makes it difficult to perform theoretical validation, such as stoichiometric and mass balance checks.

In this work, we integrated the BioCyc PGDB into the model construction process using the ScrumPy software package to generate *i*CN1361, a GSM of *C. necator* H16 that includes the database identifiers for metabolites and reactions, whilst also enabling faster and automated model refinement as the database is updated. Furthermore, the ScrumPy version of our model uses a modular framework, which separates the automated and manually curated reactions, as well as different metabolic subsystems, and thus enhances the readability of the model for future researchers to understand and use. Importantly, unlike the RehMBEL1391 model, we could demonstrate that *i*CN1361 is stoichiometrically and mass balanced.

We tested the performance of our new GSM using a more comprehensive set of experimental data. As part of this validation, we first demonstrated that the model can predict growth phenotypes for a variety of feedstocks. Additional validation, which compared the GSM results to previously published $^{13}$C metabolic flux analysis, demonstrated that *i*CN1361 is accurately predicting internal fluxes of central carbon metabolism, whilst *C. necator* H16 is growing on fructose. Notably, accurately predicting internal metabolic fluxes is fundamental to the rational design of metabolic engineering strategies for strain development.

Also important in strain development, is the ability of the model to predict metabolic behaviour during targeted engineering. In this work, we have therefore carried out a genome-wide gene essentiality screening using a TraDIS approach to determine *in vivo* knockout phenotypes for assessing *i*CN1361's predictive ability. Importantly, *i*CN1361 achieved an overall performance of 92%, precision accuracy of 81% and recall accuracy of 62% for predicting gene essentiality phenotypes. Moreover, we could improve the recall performance to 69% by integration of gene expression data to remove inactive isoenzymes. To improve the recall measure, further, incorporation of regulation, enzyme kinetics and/or thermodynamic constraints are likely required to remove alternative flux solutions in the model, which are not feasible *in vivo*. Comparing the discrepancies to Tn-Seq gene essentiality predictions available from a recent study [24], showed that 70% of the false negatives agreed with the TraDIS results, further suggesting additional constraints are required to correct these gene knockout phenotypes in *i*CN1361.

To address the discrepancies between the GSM and TraDIS gene essentiality predictions defined as false positives, a classical genetics approach was carried out to test the fitness of *C. necator* H16 mutants, carrying the in-frame deletions of 6 genes that were predicted to be essential by *i*CN1361 but not by TraDIS (or by the Tn-Seq predictions for 3 out of the 6 genes), during growth in FMM. None of these mutants grew under these experimental conditions, thus confirming the *in silico* predictions. The ambiguity of gene essentiality for genes that lie close to the determined threshold, as well as intrinsic limitations of the TraDIS-based approaches are likely responsible for these misclassified genes. Overall, our results clearly indicate that GSM and TraDIS present complementary advantages and limitations in predicting gene essentiality. We therefore propose a new "gold standard" pipeline for the identification of

conditionally essential genes in bacteria that is based on the integration of *in silico* (GSM) and *in vivo* (TraDIS and/or Tn-Seq) system-level approaches. Wherever feasible, additional data derived from transcriptomic and classical genetics analysis should also be incorporated in this pipeline to minimise the points of conflict between the *in silico* and *in vivo* genotype-phenotype relationship predictions.

Notably, useful insights from a metabolic engineering point of view can be extracted from the results. Indeed, the identification of the essential core genome of a given bacterial species represents the first step towards the construction of streamlined strains that only retain the minimal set of essential genetic functions. The use of these strains as microbial chassis for biotechnological applications should improve yields and economics of these bioprocesses since the inactivation of non-essential cellular functions is likely to result in a more efficient utilisation of resources, with respect to wild-type strains. Moreover, because of their decreased metabolic complexity, genome streamlined bacterial strains can serve as powerful tools to improve GSM predictions and, in turn, facilitate further strain development processes. Furthermore, such information can also be useful when considering suitable growth-coupling strategies, which rely on the inactivation of an essential function to force flux towards heterologous pathways for restoring growth.

Additionally, we used *i*CN1361 to investigate the metabolic changes occurring during nitrogen-limited conditions to provide insights for designing strains with improved production of PHB or, more generally, PHAs, which are co-polymers generated by the combination of 3HB with other hydroxy-alkanoates and have improved chemical and mechanical properties, as compared to PHB. We found that performing standard FBA under conditions simulating nitrogen depletion resulted in pyruvate production, as previously demonstrated *in vivo* using a PHB-negative strain [43]. Using the iMAT method, we generated a condition-specific model to represent *C. necator* H16 during nitrogen-limitation, which correctly predicted PHB production as the sole product (other than $CO_2$). Further insights from the predicted internal fluxes suggested metabolic rewiring that interestingly included the up-regulation of the CBB cycle, which balances excess electrons whilst simultaneously avoiding carbon loss *via* $CO_2$. Importantly, an active CBB cycle for growth on fructose has previously been observed experimentally but could not be predicted using the *i*CN1361 base model, due to the high level of redundancy in the model that prevented the results from converging in our analysis. Both results demonstrate the importance of solution space reduction by experimental data for a reliable analysis of biological properties using GSMs. Notably, the iMAT algorithm can also be integrated with proteomics data, and thus it would be interesting to investigate whether a proteomics-based iMAT model produces similar results. Currently, however, only transcriptomics data are available for *C. necator* H16 growing in batch culture. For continuous culture see [24].

Additionally, we predict whether metabolic activity is transcriptionally or post-translationally controlled for several important metabolic pathways, including amino acid biosynthesis and PHB metabolism. This provided valuable information that can be used for determining potential overexpression candidates to redirect metabolic fluxes towards production of a target chemical. As future work, it would be interesting to carry out dynamic modelling at the pathway level, to validate any candidates proposed from our analysis.

Whilst we have shown that *i*CN1361 can predict metabolic behaviour with a high degree of accuracy, there are many ways in which the model can be improved and developed further, which could reduce the number of discrepancies with *in vivo* results. The current model is a simplified representation of metabolism, lacking information regarding enzyme kinetics, thermodynamic feasibility, and gene expression regulation, and thus can sometimes lead to solutions that are not possible *in vivo*. Approaches such as iMAT are useful for constraining the model according to gene expression or proteomics data for a given environmental condition,

however, the approach is based on the assumption that gene expression or protein abundance is highly correlated to reaction flux, and thus still ignore the effect of kinetic and thermodynamic regulation. More advanced techniques, such as Resource Balance Analysis [24], GECKO [58] and matTFA [59], overcome these limitations by incorporating enzyme turnover rates and metabolite concentrations, to reduce the solution space. As data availability improves for *C. necator* H16, these tools can be applied to *i*CN1361 to improve the predictive accuracy of the model.

To summarise, we have presented a new GSM of *C. necator* H16, named *i*CN1361, that allows for easier reusability by other researchers, which should benefit the greater scientific community by facilitating the further development of the model as future experimental data become available. Importantly, we demonstrated that combining *i*CN1361 and TraDIS data provides an accurate platform for predicting gene knockout phenotypes in *C. necator* H16. Additionally, we showed that incorporation of omics data to overcome the lack of regulatory, kinetics and thermodynamic information in *i*CN1361, improves predictions and provides useful metabolic insights. More broadly, we expect that the model, as well as the results presented, will provide useful information for guiding engineering efforts for facilitating the implementation of *C. necator* H16 as a microbial chassis for biotechnological applications.

## Methods

### Genome-scale model construction pipeline

We used the pipeline outlined in [30,31,33] for constructing the model using the ScrumPy2.0 software package [26]. This approach uses a modular framework for building a genome-scale metabolic model, such that the model is divided into several submodels that are combined by a top-level module ('MetaReutro.spy'). This approach allows for reactions, derived automatically or manually, to be defined separately, which is helpful for managing the development and curation of the model. In the following we describe the 7 submodels that were constructed for the *i*CN1361 model.

### Extraction of BioCyc reactions for *C. necator* H16 (AutoReutro.spy)

The initial step of the construction pipeline involves the construction of a draft model based on the information provided in BioCyc's Pathway Genome Database (PGDB) for *C. necator* H16. To do this in an automated way, we downloaded the flat files of the PGDB (Reutro, v. 21.0) from BioCyc's FTP site. The module 'PyoCyc' in ScrumPy was then used to parse the information from these files to extract a draft model of reactions and metabolites. Some reactions in the PGDB include generic metabolite names, such as a 'carboxylate', an 'amino acid' and an 'aldehyde' and were either removed from the model or substituted with specific metabolite instances. The remaining reactions were checked and corrected for any stoichiometric inconsistencies in regard to C, N, S, P, O and H atoms.

### Electron Transport Chain (ETC.spy)

The electron transport chain (ETC) reactions were manually defined in the submodel ETC. spy. These reactions involve proton translocation from the cytoplasmic space to the periplasmic space, generating energy in the form of a proton gradient, which results in protons being re-consumed *via* the ATP synthase reaction. The protons in the intracellular and periplasmic space compartments have unique identifiers to avoid problems with stoichiometry. The reactions involved in the ETC were extracted from [35].

## Biomass transport reaction and ATP maintenance (Biomass.spy)

Cellular growth is represented in the model by an artificial reaction (included in Biomass.spy) that consumes the individual components, in the correct composition, that are required for producing 1 g dry cell weight of biomass. The units of the biomass precursors (defined on the left-hand side of the reaction) are in mmol·gDCW$^{-1}$. Additionally, we include the growth-associated maintenance (GAM) in the biomass reaction, as previously described in [60]. Previous work [9] has estimated GAM as 15.3 gATP·gDCW$^{-1}$, which we then converted to 30.166 mmol·gDCW$^{-1}$ and include in the left-hand-side of the biomass equation. The non-growth associated maintenance (NGAM), which has been estimated as 3.0 mmol·gDCW$^{-1}$·h$^{-1}$ in [9], is represented by an independent reaction that consumes ATP:

$$\text{ATPM} : \qquad \text{ATP} + \text{H}_2\text{O} \rightarrow \text{ADP} + \text{P}_i + \text{H}^+$$

We also include this reaction in the submodel 'Biomass.spy'.

The full list of biomass components and the calculations are provided in Table D in S1 Data). Here, we have used the data from [9] to define the compositions of the protein, DNA, RNA, phospholipids (including the fatty acids within), carbohydrate, cofactors and vitamins. The macromolecular composition provided in [9], however, includes a considerably high protein content and very low RNA content. We therefore adjusted these values based on the average *E. coli* cell that is provided in [41].

## Alternative biomass representation (BiomassTrReacs.spy)

An alternative approach for modelling biomass involves the addition of individual transport reactions for each biomass precursor. The flux through the lower and upper bounds of the reaction can then be set according to the molecular composition of the cell. This kind of representation of the biomass is useful for identifying which biomass components are growth limiting in the curation stage, and for carrying out sensitivity analysis to small changes in the biomass composition. This representation also enables variation in the biomass composition that can occur at different growth rates and for growth on different substrates.

## Phospholipids and fatty acid biosynthesis pathways (PHL.spy)

The generic pathways for phospholipid biosynthesis were extracted from the PGDB database using the PyoCyc module in ScrumPy. These pathways contain the metabolite 'an acyl-[acyl-carrier protein]' in some of the reactions, which is to account for the possibility of different fatty acids being consumed. We therefore manually added a copy of these reactions to the submodel 'PHL.spy' file for each of the fatty acids found in *C. necator*. Each reaction was given a distinct identifier by adding a suffix relating to the number of carbons in the specific fatty acid (e.g., '_C16' was added if palmitate was consumed). The BioCyc database is limited to the biosynthesis of even chain saturated fatty acids. We therefore manually added the reactions involved in the synthesis of odd chain fatty acids to this submodel. Similarly, the biosynthesis of palmitoleate was available but the other unsaturated fatty acids were manually added and included in this submodel.

## Lipopolysaccharides biosynthesis pathways (LPS.spy)

We included the generic pathway to lipopolysaccharide (LPS) biosynthesis available in the MetaCyc PGDB (v 21.0) for *E. coli* in a separate submodel. The LPS composition can vary in different bacteria, therefore including it as a separate submodel allows easy curation of the

reactions. To our knowledge, the LPS composition has currently not been defined for *C. necator*, and so was assumed to be the same as the LPS composition in *E. coli* in *i*CN1361.

## Gap filling (ExtraReacs.spy)

Due to the incompleteness of the BioCyc PGDB, essential reactions for producing the biomass precursors were absent from the model. These missing reactions were identified *via* Blast+ or the KEGG database and added to the ExtraReacs.spy module. Similarly, gap-filling was carried out to fulfil growth in the model on feedstocks for which experimental evidence exists for *C. necator* H16.

## Transport reactions for uptake and secretion of metabolites (Transporters. spy)

A transport reaction for each metabolite known to be consumed or produced by *C. necator* H16 was added to a separate submodel. These reactions all include the suffix '_tx', which makes them easily identifiable for setting constraints on uptake and production. The prefix 'x_' was also added to metabolites that were in the extracellular compartment. Notably, we also represent PHB storage using an artificial transport reaction in this submodel.

## Gene-reaction relationships

The gene-reaction relationships are defined in a GSM using boolean logic [60,61]. A reaction that is associated with an enzyme complex is dependent on all genes within the complex being expressed and are therefore represented by an 'AND' relationship. Alternatively, there exist some reactions that are associated with multiple genes (*i.e.*, isoenzymes), and rely on only 1 of the genes being active. Reactions associated with isoenzymes are represented using an 'OR' relationship. To identify the gene-relationships for *C. necator*, we first used the PyoCyc module in ScrumPy to automatically derive a list of genes associated to each reaction. The genes associated with a reaction were defined as a complex only if they were present in the same operon. If the genes are being expressed simultaneously then it is highly likely they form a complex. The information regarding which operon a gene was contained in, was also automatically extracted from the PGDB database using the PyoCyc module in ScrumPy. The remaining genes were then assumed to be acting as isoenzymes. Any reaction that had been added to the model through the initial automatic construction but was not essential for a known metabolic function and for which a gene association could not be determined, was removed from the model. Missing gene annotation for essential reactions were checked in KEGG and UniProt and added to the model if available.

## Model analysis

The set of reactions in the GSM are represented mathematically using a stoichiometric matrix, $N$, in which rows represent the $m$ metabolites and columns represent the $n$ reactions. The entries in $N$ correspond to the stoichiometries of the metabolites in the reaction. Negative and positive entries are used to represent metabolite consumption and production in the reaction, respectively. The changes of metabolite concentrations can be modelled using a system of kinetic equations, such that

$$\frac{d\boldsymbol{x}}{dt} = N\boldsymbol{v} \tag{1}$$

where $\boldsymbol{v}$ is a vector of length $n$ corresponding to the fluxes through reactions and $x$ is a vector

of metabolite concentrations of length $m$. Eq 1 can then be simplified by assuming the metabolite concentrations have reached a state where they are not changing over time ($N\boldsymbol{v} = 0$). For most metabolic networks, the number of reactions exceeds the number of metabolites ($n>m$), and results in multiple solutions existing for this coupled system of equations. Constraint based approaches can be applied to explore possible flux distributions.

## Flux balance analysis (FBA)

In this work, we identified feasible flux distributions using flux balance analysis (FBA) [62]. FBA finds the optimal flux distribution that maximises or minimises some biologically relevant objective function, $Z$, whilst also satisfying a set of constraints. The objective function can be any linear combination of fluxes multiplied by a vector of weights ($Z = \boldsymbol{c}^T\boldsymbol{v}$). Additional constraints are added to the LP to reduce the feasible solution space. Knowledge of reaction directionality and metabolite uptake and production rates are incorporated into the LP as lower and upper bounds on reactions ($\boldsymbol{v}_{lb}$, $\boldsymbol{v}_{ub}$), for example. The LP is formulated as follows:

$$
\begin{aligned}
\text{max.} \quad & Z = \boldsymbol{c}^T\boldsymbol{v} \\
\text{s.t.} \quad & N\boldsymbol{v} = 0 \\
& \boldsymbol{v}_{lb} \leq \boldsymbol{v} \leq \boldsymbol{v}_{ub}
\end{aligned}
\tag{2}
$$

In this work, we assumed that bacteria evolve to optimise their cellular growth, whilst also minimising their enzymatic burden. This method is called parsimonious flux balance analysis (pFBA) and involves solving two LPs [63]. Standard FBA is performed in the first step with maximisation of the growth rate (*i.e.*, the flux towards the biomass equation) as the objective function. The optimal growth rate is then fixed as an additional constraint. The second LP is then solved with minimisation of the total sum of fluxes as the new objective function:

$$
\begin{aligned}
\text{min.} \quad & |\boldsymbol{v}| \\
\text{s.t.} \quad & N\boldsymbol{v} = 0 \\
& \boldsymbol{v}_{lb} \leq \boldsymbol{v} \leq \boldsymbol{v}_{ub} \\
& \boldsymbol{c}^T\boldsymbol{v} = \boldsymbol{Z}
\end{aligned}
\tag{3}
$$

where $\boldsymbol{c}$ contains a weight of 1 in the position corresponding to the biomass reaction and zero elsewhere. To mimic minimal media in aerobic conditions, we constrained both LPs to allow for the free uptake of sulfate, phosphate, ammonium, oxygen, water and protons. All other transport reactions had their lower bounds set to zero.

## Testing biomass precursor production

Throughout the curation stage, we tested whether the model could synthesise each precursor in the biomass composition. To test an individual precursor, we added an artificial transport reaction that consumed the metabolite. The LP in Eq (2) was then modified to maximise this transport reaction as the objective function. If no solution existed, then the model was inspected for incorrect reaction directionality and/or gaps in the network. To check the reaction directions, we re-ran the LP but with all reactions set as reversible. If a solution to the LP is now feasible, then the set of reactions carrying a negative flux, which were previously constrained as irreversible, are selected as candidates for further curation. Each candidate was then checked in BRENDA [28] and eQuilibrilator [64] for any evidence to suggest the reaction is reversible.

### Theoretical validation

We checked that the model obeyed the law of mass and energy conservation. First, we verified that the model was not capable of synthesising energy-equivalents without the input of electrons. To do this, we modified the LP in Eq (2), to maximise the ATP maintenance reaction as the objective function, whilst constraining the transport of all carbon and energy sources to zero. If a solution exists, then reactions were checked for inconsistencies in their stoichiometry and their directionality. Second, we also computed the set of unconserved metabolites using ScrumPy [65]. Any reaction that involves an unconserved metabolite requires rechecking for a stoichiometric imbalance. We provide a MEMOTE report [66] in the supplementary material (https://github.com/SBRCNottingham/CnecatorGSM/tree/main/MEMOTE_repo/sbrc_cnecator_gsm), which shows the results of these consistency checks.

### Carbon source utilisation simulations

For this analysis, a simple transport reaction that consumed the carbon source under study was added to the model with a lower bound fixed at 10 mmol·gDCW$^{-1}$·h$^{-1}$. The uptake rate of all other carbon sources in the model were fixed to zero. Maximisation of the biomass equation was then set as the objective in the pFBA simulation (Eq 2). If a feasible solution was found with a growth rate greater than 0.01 h$^{-1}$, then the carbon source was considered as growth supporting.

### GSM gene essentiality analysis

A gene deletion was simulated in the model by constraining the flux through any associated reaction, to which no alternative isoenzyme could be utilised, to zero. pFBA was then re-run to simulate the growth of the knockout mutant. We then calculated the 'importance' of a gene by dividing the growth rate predicted for the knockout mutant by the growth rate predicted for the wild type under the same conditions. We considered a gene as essential for biomass production if its deactivation in the model results in a growth rate of less than 0.05 h$^{-1}$.

### Integration of transcriptomics data

Gene expression data was integrated with the GSM using the method iMAT [45]. To run iMAT, the reactions in the model are first categorised into three divisions: high, moderate and low expression for each condition of interest. Reaction expression was determined by taking the total reads per kilobase per million (RPKM) of the gene(s) that encodes the enzyme for catalysing that reaction. The following rules were applied for reactions with a complex gene-reaction relationship:

1. Reactions with 'OR' GR-relationships (*i.e.*, isoenzymes): the reaction expression was determined by summing the RPKM value of all isoenzymes.

2. Reactions with 'AND' GR-relationships (*i.e.*, multi-subunit enzyme): the reaction expression was determined using the minimum RPKM value of all genes in the complex.

3. Reactions with 'AND' and 'OR' GR-relationships: the reaction expression was calculated by combining the rules from (1) and (2).

The reaction expression was integrated into the GSM using iMAT for the f16 condition. For the f26 condition, we considered both the absolute reaction expression and the fold change by multiplying the two together. By doing so, we give a high weighting to reactions that have a high absolute expression and/or high fold change between conditions. Reactions that were

associated to genes with an RPKM greater than the 95[th] percentile were associated with high activity, similar to the approach carried out in [46]. To determine the low expression cut-off, we identified the minimal expression value (excluding outliers) for the reactions that are essential in the model, which guarantees that all essential reactions remain active in the new solution. iMAT was then used, which applies a mixed-integer linear program (MILP), to maximise the number of active reactions in the high category, whilst simultaneously minimising the number of active reactions in the low category. This approach was applied to produce condition-specific models for growth during the growth phase (f16) and nitrogen depleted phase (f26). Note that the maximum growth rate specific for each phase (f16 and f26) was used to constrain the model before generating the two condition-specific models. These growth rates were calculated by fitting a logistic growth model and straight line to the biomass curves corresponding to the growth phase and nitrogen-depleted phase, respectively, which are provided in [42] (see Table A in S5 Data).

## Identifying differentially altered reactions

Flux sampling was used to identify reactions whose flux was significantly altered between the two conditions. Flux sampling was carried out on each condition-specific iMAT model using the optGpSampler sampling method [44], which is available in the cobrapy toolbox. A thinning factor of 1000 was applied and 10,000 sample points for each model were returned from the solution space. Flux sampling was run twice for each model to ensure that the 10,000 samples were a good representation of the solution space. The two-sample Kolmogorov-Smirnov (KS Test) was used to test whether the distribution was the same for each reaction across both sampling runs. Any reaction whose flux distribution was significantly different ($p < 0.05$) was removed from any further analysis. The approach used in [47] was then applied to filter the reactions that had significantly increased/decreased flux between the two conditions. Here, the two-sample KS Test is again used with a significance level of 0.05 for determining whether the flux distributions for each reaction are different. The p-values were adjusted to account for multiple testing using the Benjamini-Hochberg FDR correction using a significance level of 0.05 [47]. The mean flux of the 10,000 samples for each reaction is then calculated for each condition. The normalised flux change (FC) is calculated as the following, as proposed in [47]:

$$\text{Flux change (FC)} = \frac{\bar{S}_{f26} - \bar{S}_{f16}}{\bar{S}_{f26} + \bar{S}_{f16}}$$

where $\bar{S}$ represents the mean of the distribution for each reaction for the corresponding condition. A FC greater than 0.33 (corresponding to a fold change of 2) was used as the threshold for filtering out significantly altered reactions. Bootstrapping was used to estimate the 95% confidence interval for reactions that were present in only one model, which was also applied in [47]. A reaction is then considered differentially altered if zero is outside the 95[th] confidence interval.

## Bacterial strains and growth conditions

All the bacterial strains and plasmids used in this study are listed in Table A in S6 Data. Plasmid vectors, together with their nucleotide sequences, may be sourced from www.plasmidvectors.com. Standard lysogeny broth (LB) was used for general maintenance of *C. necator* and *E. coli* strains. Low-salt-LB (LSLB)-MOPS medium [67] was used when growing *C. necator* H16 as recipient in conjugative procedures and Hanahan's Broth (SOB Medium—H8032, Sigma-Aldrich) for the preparation of *C. necator* H16 competent cells and for the

TraDIS experiment. Chemically defined minimal medium (MM) [68] was supplemented with 4 g/l of either D-fructose (FMM), when used to grow the *C. necator* H16 transposon mutant library for TraDIS, or sodium D-gluconate (SGMM), when used to select *C. necator* H16 transconjugants. When needed, antibiotics were added to the medium at the following concentrations: 10 μg/mL gentamycin, 50 μg/mL chloramphenicol or 15 μg/mL tetracycline. All the antibiotics and chemicals used in this study were purchased from Sigma-Aldrich. Unless stated otherwise, *E. coli* and *C. necator* strains were grown aerobically in a shaking incubator (Thermo Scientifc MaxQ 8000 Incubated Stackable Shaker) at 37 and 30˚C, respectively, with shaking at 200 rpm.

## Construction of a *C. necator* H16 transposon mutant library

To generate a transposon mutant library in *C. necator* H16, a suicide vector (pMTL70115) carrying a miniTn5 transposon-delivery system was designed and constructed. The *tnpA* gene, encoding a hyperactive transposase and a miniTn5 transposon, carrying a tetracycline-resistance gene, were obtained as synthetic DNA constructs (GeneArt–Invitrogen). The DNA sequences of *tnpA* and the transposon mosaic ends (MEs) were derived from the pBAM vector series [69,70], while the sequence of the *tetA* tetracycline-resistance gene was obtained from plasmid pBBR1MCS-3 [71]. The DNA parts carrying the *tnpA* transposase-encoding gene and the miniTn5::*tetA* transposon were digested using the restriction endonucleases (REs) PstI-HF/SpeI and SpeI/FseI, respectively. These constructs were then cloned together into the backbone of a modular vector belonging to the pMTL70000 series [72], harbouring the *catP* chloramphenicol-resistance gene and the ColE1 origin of replication from plasmid pMTL20 [73], which was previously digested with the REs PstI-HF/FseI. All the REs used in this study were purchased from New England Biolabs (NEB). The resulting plasmid (pMTL70115) was propagated and purified from *E. coli* strain C2925 (NEB), which is defective for both Dam and Dcm DNA methylation, prior to being transformed into *C. necator* H16. This was done to increase transformation and transposition efficiencies, since *C. necator* H16 encodes a Type IV restriction-modification (R-M) system capable of degrading invading methylated DNA that was shown to negatively affect plasmid DNA transformation efficiencies in this species [74]. Competent cell preparation and electroporation of *C. necator* H16 were performed as described previously [75]. Briefly, electroporation was carried out using 500 ng of pMTL70115 transposon delivery vector. Following 4 h of recovery in 1 ml of SOB Medium at 30˚C with shaking, the transformation mixtures were plated on LB agar supplemented with 15 μg/ml tetracycline and 5 mM MgSO$_4$. This process was repeated several times to select a total of approximately 1,032,000 *C. necator* H16 transposon mutants. In addition, a total of 500 colonies were randomly picked from independent transformations and replica-plated on LB agar supplemented with 15 μg/ml tetracycline, in the presence or absence of 50 μg/ml chloramphenicol to confirm curing of plasmid pMTL70115. Integration of the miniTn5::*tetA* transposon in the genome of the tetracycline-resistant, chloramphenicol-sensitive mutants was confirmed by PCR with primers MCSTn5_FOR and Tn5_NCOseq3_REV (see Table B in S6 Data, using Green GoTaq DNA polymerase 2x Master Mix (Promega). The *C. necator* H16 transposon mutants were scraped off the selection plates with plastic loops, resuspended in a sterile solution of 20% (v/v) glycerol in PBS and aliquoted in seventy-seven 2 ml sub-pools, each containing approximately 15,000 colonies, before being stored at -80˚C.

## TraDIS experiments set up

To set up TraDIS, 20 μl aliquots were taken from each of the seventy-seven sub-pools of *C. necator* H16 transposon mutants, combined and inoculated (380 μl of transposon mutant

library per culture) in 100 ml Erlenmeyer flasks filled with 20 ml of either SOB medium or FMM. Two independent cultures were set up for each condition and flasks were incubated at 30˚C, with shaking. FMM cultures were cultivated for five growth passages, by re-inoculating 1 ml of culture in 19 ml of fresh FMM every 24 h, while SOB cultures were stopped after 24 h of incubation. Three 6 ml samples were collected from each culture and genomic DNA was purified, using the GenElute Bacterial Genomic DNA kit (Sigma-Aldrich).

### Preparation of DNA samples for sequencing

Genomic DNA was fragmented using Covaris sonication to an average length of approximately 300 bp. The fragments were then end-repaired and prepared for adapter ligation with the NEB ultra II kit library preparation kit. TruSeq adapters were ligated onto the ends of fragments to allow an amplicon library to be created using a P5 primer which binds to the transposon and a P7 primer which binds the adapter. PCR products were run on a low-melt agarose gel and then gel extracted. These gel-purified products were analysed on an Agilent DNA Bioanalyzer chip and a qPCR was performed to determine appropriate quantities of DNA to apply to the Illumina flowcell. Two libraries were generated for each sample (one for each end of the transposon insertion cassette) and then all samples were multiplexed. Illumina MiSeq runs were performed using a 15% PhiX spike to provide sufficient diversity for the first 10 reads, to prevent run failure. Confirmed TraDIS reads were identified and separated by the presence of the 11 bp sequence (ATAAGAGACAG) that flanks either one of the two MEs of the transposon cassette, at the 5' end of the reads. These sequences were trimmed from the reads using the python package cutadapt (https://pypi.python.org/pypi/cutadapt), allowing for 20% mismatch. The remaining non-specific reads without adapter that were generated as a by-product of the library prep were discarded. TraDIS reads were further trimmed for sequencing adapters also using cutadapt. Trimmed reads were mapped to the *C. necator* H16 genome using BWA MEM [76].

### Mapping homologues in *Cupriavidus necator* H16 to *Burkholderia cenocepacia* H111

We downloaded the Genbank files for the replicons of *C. necator* H16 (Chromosome 1: AM260479.1, Chromosome 2 AM260480.1 and Megaplasmid pHG1 AY305378.1; annotation date 07/03/2015 and 25/07/2016 for the megaplasmid, respectively) from the Genbank website and *B. cenocepacia* H111 (Chromosome 1: NZ_HG938370.1, Chromosome 2: NZ_HG938371.1 and Chromosome 3: NZ_HG938372.1; annotation date 13/12/2020) from the NCBI RefSeq website. Then, we extracted the coding sequences for the proteins for both organisms. Note that we have included sequences of pseudogenes because functional homologues may exist in the other species. The set of coding sequences were then used to create separate Blast databases representing the whole genome of each organism. Finally, each coding sequence was blasted against the other organism database in an automated procedure identifying the best matching protein. The BLAST+ 2.11.0 software package was used for these two steps. The resulting mappings are listed in files Map_Bcen_2_Cnec_H16.xlsx and Map_Cnec_H16_2_Bcen.xlsx (available via https://github.com/SBRCNottingham/CnecatorGSM/upload/main), including the proposed function of the homologues and statistical information about the hit, i.e., e-value, identity, and coverage.

### Experimental determination of essential genes using TraDIS

The number of insertions per kilobase per million (IPKM) was calculated for each CDS, as previously described [40], for both chemically defined (FMM) and complex (SOB) media conditions. These values were then curated by discarding the transposon insertions that mapped

within the first 5% (5'-end) and the last 20% (3'-end) of each CDS, as previously reported in [77], to obtain the IPKM_curated (IPKMc) values. A set of genes that was recently experimentally validated as essential in the closely related bacterial species *Burkholderia cenocepacia* H111, for growth in LB medium [77], were used for determining an IPKMc threshold for classifying the *C. necator* H16 genes as essential or non-essential. The genes in *C. necator* H16 were mapped to these essential genes from *B. cenocepacia* H111 (see previous section) and those that showed a high degree of homology ($p < e^{-50}$) were subsequently used for determining the threshold. Note that only the homologous genes involved in DNA replication, transcription and translation, protein modification, protein transport and cell cycle, which have been identified as common essential processes across bacterial species [29,77–79], were considered to minimise bias from any differences in the metabolisms of the two bacterial species. The 65th percentile of the log-distribution of the SOB-specific IPKM values ($\log_2(IPKMc+1)$) was then selected as the threshold to classify the *C. necator* H16 genes as essential ($\log_2(IPKMc+1) \leq$ lower threshold), whereas the 75th percentile was used to classify genes as non-essential ($\log_2(IPKMc+1) >$ upper threshold). A gene that had a $\log_2(IPKMc+1)$ index within the 65th and 75th percentiles was not classified as essential or non-essential in our analysis. The following three performance metrics were calculated to test the accuracy of the GSM predictions compared to the TraDIS results, such that P and N indicate essential and non-essential cases respectively, T indicates true (correct) and F false (wrong) predictions:

- Recall (TPR—true positive rate) = TP/(TP + FN)

- Precision (PPV—positive predictive value) = TP/(TP + FP)

- Accuracy (ACC) = (TP + TN)/(P + N)

## Software

The model construction and theoretical validation was all ran using ScrumPy2.0 (http://mudshark.brookes.ac.uk/ScrumPy) [26]. Model consistency checks were ran using MEMOTE v0.13.0 [66]. Model analysis was carried out using the COBRApy toolbox [80]. Condition-specific models were constructed using the COBRA toolbox [81] in Matlab R2018a. All simulations were ran using the GNU Linear Programming Kit (GLPK) solver.

## Supporting information

**S1 Data. (Tables A-E): List of reactions; list of metabolites; definitions of the gene-reaction relationships and the list of inconsistent enzyme subsets in *i*CN1361.**
(XLSX)

**S2 Data. The predicted growth phenotypes for *i*CN1361 for 131 different carbon sources.**
(XLSX)

**S3 Data. The flux values predicted using parsimonious FBA and flux variability analysis using *i*CN1361 and compared to $^{13}$C metabolic flux analysis.**
(XLSX)

**S4 Data.** (Tables A-F): Derivation of the $\log_2(IPKMc+1)$ threshold for classifying gene essentiality in the TraDIS results; comparison of *in vivo* (TraDIS) and *in silico* (*i*CN1361) gene essentiality phenotypes; list of *i*CN1361's false positive (essential) genes and list of *i*CN1361's false negative (non-essential) genes; Numbers of transposon insertions and average insertion frequencies observed for each of the three *C. necator* H16 replicons (chromosomes 1 & 2 and the pHG1 megaplasmid) following growth in SOB rich medium and fructose mineral

medium (FMM).
(XLSX)

**S5 Data.** (Tables A-C): Derivation of growth rates from experimental data; results from the flux sampling simulations for the *i*CN1361-f16 and *i*CN1361-f26 condition-specific models and predicted regulation of reaction activity for 14 amino acid biosynthetic pathways compared to regulation in EcoCyc.
(XLSX)

**S6 Data.** (Tables A-B): List of bacterial strains and plasmids and list of oligonucleotide primers used in this study.
(DOCX)

## Acknowledgments

We would like to thank the BBSRC NIBB, C1net for organising the C1net conferences and workshops, which facilitated the collaborations of the University of Nottingham with Oxford Brookes University and with the Korean Advanced Institute of Science and Technology.

## Author Contributions

**Conceptualization:** Nicole Pearcy, Marco Garavaglia, Thomas Millat, James P. Gilbert, David A. Fell, Mark Poolman, Jamie Twycross, Nigel P. Minton.

**Formal analysis:** Nicole Pearcy, Marco Garavaglia, Thomas Millat, Yoseb Song, Hassan Hartman, Craig Woods, David A. Fell, Klaus Winzer.

**Funding acquisition:** John R. King, Klaus Winzer, Jamie Twycross, Nigel P. Minton.

**Investigation:** Nicole Pearcy, Marco Garavaglia, Thomas Millat, Rajesh Reddy Bommareddy.

**Methodology:** Nicole Pearcy, Marco Garavaglia, Thomas Millat, Hassan Hartman, Claudio Tomi-Andrino, David A. Fell, Mark Poolman, Jamie Twycross.

**Project administration:** John R. King, Klaus Winzer, Jamie Twycross, Nigel P. Minton.

**Software:** Nicole Pearcy.

**Supervision:** Thomas Millat, Hassan Hartman, Byung-Kwan Cho, David A. Fell, Mark Poolman, John R. King, Klaus Winzer, Jamie Twycross, Nigel P. Minton.

**Validation:** Marco Garavaglia.

**Visualization:** Nicole Pearcy, Marco Garavaglia.

**Writing – original draft:** Nicole Pearcy, Marco Garavaglia, Thomas Millat, James P. Gilbert, Craig Woods.

**Writing – review & editing:** Nicole Pearcy, Marco Garavaglia, Thomas Millat, James P. Gilbert, Yoseb Song, Hassan Hartman, Craig Woods, Claudio Tomi-Andrino, Rajesh Reddy Bommareddy, Byung-Kwan Cho, David A. Fell, Mark Poolman, John R. King, Klaus Winzer, Jamie Twycross, Nigel P. Minton.

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
