## [Decision Letter · Decision Letter 0]

23 Dec 2021

Dear Dr Pearcy,

Thank you very much for submitting your manuscript "A genome-scale metabolic model of Cupriavidus necator H16 integrated with TraDIS and transcriptomics data reveals metabolic insights for biotechnological applications" for consideration at PLOS Computational Biology.

As with all papers reviewed by the journal, your manuscript was reviewed by members of the editorial board and by several independent reviewers. In light of the reviews (below this email), we would like to invite the resubmission of a significantly-revised version that takes into account the reviewers' comments. Both reviewers appreciated the topic and provided important comments in their reviews. The authors are asked to address the criticisms raised by Reviewer #1.

We cannot make any decision about publication until we have seen the revised manuscript and your response to the reviewers' comments. Your revised manuscript is also likely to be sent to reviewers for further evaluation.

Sincerely,

Costas D. Maranas

Associate Editor

PLOS Computational Biology

Jason Papin

Editor-in-Chief

PLOS Computational Biology

Reviewer's Responses to Questions

**Comments to the Authors:**

Reviewer #1: Summary

In this paper, the authors reconstructed a genome-scale metabolic model (GSM) for Cupriavidus necator H16 using metabolites/reactions IDs from BioCyc database. The model is validated against multiple experimental data such as growth and 13C metabolic fluxes. The authors then performed TraDIS experiments to assess gene essentiality and compared with GSM predictions, and false positive essential genes were further verified experimentally, which revealed that some TraDIS limitations can be complemented by GSM predictions. The GSM is then integrated with transcriptomics data to build two condition-specific models. Flux sampling was used to predict flux changes during nitrogen limitation condition, and another computational approach was used to predict transcriptional or post-translational regulations. Overall, the experimental approaches and analysis are very thorough, and the paper is well-written. The major concern I had is that the authors were benchmarking their model against an old RehMBEL1391 model while an updated model [1] was published recently. A few comments are listed as follows for the authors to consider to strengthen their manuscript.

Comments

Major

- Regarding the model reconstruction, it would be more helpful to the readers if the authors can provide some comparison with the published models. For example, iCN1361 has 1292 reactions, while the number of reactions in the first RehMBEL1391 model is 1391 and the recent RehMBEL1391 model is 1360. Can the authors add some explanation on why certain reactions are added/removed? A good example is in the readme of the updated iCN1361 repos that include a list of changes made compared to the previous model (https://github.com/m-jahn/genome-scale-models).

- Biomass reaction might also need to be updated. It looks like that biomass reaction still contains pyridoxine as the required cofactor instead of pyridoxal-5-phosphate (pydx5p) as corrected by [1].

- The latest RehMBEL1391 study [1] performed gene essentiality experiments but used a different experimental approach (Tn-seq). It also included gene essentiality data from formate/succinate carbon source experiments. It is recommended to include these datasets to test the GSM predictions as well. Since the experimental approach to determine gene essentiality is different, any limitations of TraDIS mentioned by the authors in pages 17-19 did not shown up in the Tn-seq study?

- Page 25: iMAT can use transcriptomics data or proteomics data to build condition specific models. Since genome-wide proteomics data is also available in the latest study [1], the authors may want to compare the model they built with proteomic-based iMAT model to find any discrepancies.

- Page 28 lines 627-639: The conclusions the authors made about CBB cycle and CO2 fixation are different from the analysis of [1]. The conclusion from the latest RehMBEL1391 is that “CO2-reassimilation through Rubisco does not provide a fitness benefit for heterotrophic growth, but is rather an investment in readiness for autotrophy”. Can the authors provide a more detailed analysis to explain the different predictions made by the two models?

- In the proteomics data from [1], phaC enzyme level is low compared with phaA or phaB. Is it possible that the feedback inhibition is not the major limitation in f16 condition but the enzyme level of phaC? In other words, is it possible for the computational approach/analysis to be done at the pathway level to make sure that feedback inhibition is the only reason as suggested by the author?

- Page 8 lines 184-185: It is great that the authors are using MEMOTE for validation tests and generated a report in the github repository. However, the github repository is not organized using the memote format. For example, there should be a data folder to store the gene essentiality, growth, and media data. Moreover, the repository can be used to track model changes since new experimental results are often leading the model improvement [2]. It is therefore recommended that the authors follow the same standard to create a memote generated structure. A good example is https://github.com/maranasgroup/iRhto_memote

Minor

- Page 6 lines 135 - 148: It is possible that some metabolites/reactions in C. necator H16 do not exist in BioCyc. Therefore, it might make more sense to use a more general identifier (e.g., metanetx ids). This will allow the model to use information from other comprehensive databases (e.g., KEGG or Rhea) that are not added into biocyc. Can the authors comment on whether they have encountered this scenario during their model reconstruction?

- Page44 formulation (3): the objective function should be minimizing of 1-norm of flux vector.

- There are still 25 unbounded reactions based on memote report, can the authors verify that they are not forming thermodynamically infeasible cycles?

References

[1] Jahn, Michael, et al. "Protein allocation and utilization in the versatile chemolithoautotroph Cupriavidus necator." bioRxiv (2021).

[2] Dinh, Hoang V., et al. "A comprehensive genome-scale model for Rhodosporidium toruloides IFO0880 accounting for functional genomics and phenotypic data." Metabolic engineering communications 9 (2019): e00101.

Reviewer #2: The manuscript presents an exemplary work on genome-scale metabolic reconstruction. The draft model from an automated pipeline was curated extensively with consistency tested thoroughly. The model was validated against growth physiological data, 13-C MFA data, transposon mutant library for gene essentiality prediction and phenotypes of particular knockout mutants under questions. Transcriptomics data were then used to identify potential interactions between different pathways and regulatory control. Overall, the model captures the experimental data very well and generates insight into the metabolism of the organism being studied. This will be a useful tool for predicting metabolic engineering strategies for Cupriavidus necator H16.

Minor comment:

Only two minor comments:

- When analyzing the correlation between different pathways with the integration of transcriptomic data (e.g., RuBisCo, ED pathway, TCA cycle and PHB). Are those correlations truly the results of integrating the transcriptomic data or are they possibly just the network properties of the model? In other words, are the correlations significant compared to the base model not with transcriptomic data integrated? Comparing this might give a clearer picture.

- Is the manually curated PGDB in BioCyc format available online? This will help a lot to share the results especially to non-computational people.

**Have the authors made all data and (if applicable) computational code underlying the findings in their manuscript fully available?**

Reviewer #1: Yes

Reviewer #2: Yes

PLOS authors have the option to publish the peer review history of their article (what does this mean?). If published, this will include your full peer review and any attached files.

Reviewer #1: No

Reviewer #2: No
---

## [Decision Letter · Decision Letter 1]

14 Apr 2022

Dear Dr Pearcy,

We are pleased to inform you that your manuscript 'A genome-scale metabolic model of Cupriavidus necator H16 integrated with TraDIS and transcriptomics data reveals metabolic insights for biotechnological applications' has been provisionally accepted for publication in PLOS Computational Biology.

Best regards,

Jason Papin

Editor-in-Chief

PLOS Computational Biology

Reviewer's Responses to Questions

**Comments to the Authors:**

Reviewer #1: The authors have addressed all of my comments in details. In particular, the authors have included the comparison with a recently published Cupriavidus necator H16 model and updated their model/analysis using the new biomass, gene essentiality, and other data in the published model. I recommend the paper to be accepted to PLOS Computational Biology journal.

Reviewer #2: The authors have satisfactorily addressed the comments.

**Have the authors made all data and (if applicable) computational code underlying the findings in their manuscript fully available?**

Reviewer #1: Yes

Reviewer #2: Yes

PLOS authors have the option to publish the peer review history of their article (what does this mean?). If published, this will include your full peer review and any attached files.

Reviewer #1: No

Reviewer #2: No

---

## [Editor Report · Acceptance letter]

15 May 2022

PCOMPBIOL-D-21-02005R1 

A genome-scale metabolic model of Cupriavidus necator H16 integrated with TraDIS and transcriptomics data reveals metabolic insights for biotechnological applications

Dear Dr Pearcy,

I am pleased to inform you that your manuscript has been formally accepted for publication in PLOS Computational Biology. Your manuscript is now with our production department and you will be notified of the publication date in due course.

With kind regards,

Olena Szabo
